# Mechanisms that regulate the C1-C2B mutual inhibition control functional switch of UNC-13

**Haowen Liu[1,2†], Lei Li[1,2†], Jiafan Wang[1], Jiayi Hu[1], Jingyao Xia[3], Xiaochun Yu[1], Jing Tang[4], Huisheng Liu[5], Xiaofei Yang[6], Cong Ma[7], Lijun Kang[8], Zhitao Hu[1*]**

[1]Department of Neuroscience, City University of Hong Kong, Kowloon, China; [2]Centre for Regenerative Medicine and Health, Hong Kong Institute of Science and Innovation, Chinese Academy of Sciences, Hong Kong, China; [3]Queensland Brain Institute, The University of Queensland, Brisbane, Australia; [4]Neuroscience Research Center, Key Laboratory of Biomedical Information Engineering of Ministry of Education,School of Life Science and Technology, Xi'an Jiaotong University, Xi'an, China; [5]School of Biomedical Engineering, Guangzhou Medical University; Guangzhou National Laboratory; Bioland Laboratory, Guangzhou, China; [6]Key Laboratory of Cognitive Science, Hubei Key Laboratory of Medical Information Analysis and Tumor Diagnosis & Treatment, Laboratory of Membrane Ion Channels and Medicine, College of Biomedical Engineering, South-Central Minzu University, Wuhan, China; [7]Key Laboratory of Molecular Biophysics of the Ministry of Education, College of Life Science and Technology, Huazhong University of Science and Technology, Wuhan, China; [8]Department of Neurology of the Fourth Affiliated Hospital and School of Brain Science and Brain Medicine, Zhejiang University School of Medicine, Yiwu, China

**\*For correspondence:**
zhitaohu@cityu.edu.hk

[†]These authors contributed equally to this work

**Competing interest:** The authors declare that no competing interests exist.

## eLife Assessment

This **important** study by Liu et al. presents a comprehensive structure-function analysis of the presynaptic protein UNC-13, leading to new insights into how its distinct domains control neurotransmitter release. The methods, data, and analyses are **convincing**, and the genetic and electrophysiological approaches support many of their conclusions. The work will be of interest to neuroscientists studying synaptic transmission, as it provides a foundation for future mechanistic studies of Munc13/UNC-13 family proteins.

**Abstract** Munc13 plays a crucial role in short-term synaptic plasticity by regulating synaptic vesicle (SV) exocytosis and neurotransmitter release at the presynaptic terminals. However, the intricate mechanisms governing these processes have remained elusive due to the presence of multiple functional domains within Munc13, each playing distinct roles in neurotransmitter release. Here, we report a coordinated mechanism in the *Caenorhabditis elegans* Munc13 homolog UNC-13 that controls the functional switch of UNC-13 during synaptic transmission. Mutations disrupting the interactions of C1 and C2B with diacylglycerol (DAG) and phosphatidylinositol 4,5-bisphosphate (PIP$_2$) on the plasma membrane induced the gain-of-function state of UNC-13L, the long UNC-13 isoform, resulting in enhanced SV release. Concurrent mutations in both domains counteracted this enhancement, highlighting the functional interdependence of C1 and C2B. Intriguingly, the individual C1 and C2B domains exhibited significantly stronger facilitation of SV release compared

to the presence of both domains, supporting a mutual inhibition of C1 and C2B under basal conditions. Moreover, the N-terminal C2A and X domains exhibited opposite regulation on the functional switch of UNC-13L. Furthermore, we identified the polybasic motif in the C2B domain that facilitates SV release. Finally, we found that disruption of C1 and C2B membrane interaction in UNC-13S, the short isoform, leads to functional switch between gain-of-function and loss-of-function. Collectively, our findings provide a novel mechanism for SV exocytosis wherein UNC-13 undergoes functional switches through the coordination of its major domains, thereby regulating synaptic transmission and short-term synaptic plasticity.

## Introduction

Synapses exhibit many types of short-term, use-dependent synaptic plasticity which are thought to underlie learning and memory formation (*Fioravante and Regehr, 2011*; *Regehr, 2012*). The regulatory mechanisms of synaptic plasticity involve the functional diversity of the key synaptic proteins in synaptic vesicle (SV) release (*Jahn and Südhof, 1999*; *Südhof and Rizo, 2011*). Exocytosis of SVs releases neurotransmitters to transfer neuronal signals, and this membrane fusion process is controlled by an exquisite release machinery, consisting of the core SNARE complexes including SNAP-25, syntaxin, and synaptobrevin, and SNARE-interacting proteins such as Munc13, complexin, and Munc18 (*Jahn and Südhof, 1999*; *Rizo and Rosenmund, 2008*; *Südhof and Rizo, 2011*). Previous studies have shown that many synaptic proteins in the release machinery contain multiple functional domains playing different roles in regulating SV release (*Martin et al., 2011*; *Neher, 2010*; *Nishiki and Augustine, 2004*; *Weber et al., 2010*; *Xu et al., 2009*; *Xue et al., 2009*). Deciphering how synaptic proteins integrate their domain functions to control synaptic strength is thus crucial for understanding the molecular mechanisms governing synaptic transmission and synaptic plasticity.

The multi-domain protein Munc13 serves as a key regulator of SV priming and exocytosis in the mammalian central synapses. Genetic knockout of Munc13, or its invertebrate homolog UNC-13, eliminates nearly all neurotransmitter release (*Aravamudan et al., 1999*; *Augustin et al., 1999*; *Richmond et al., 1999*). Recent investigations into Munc13 and its *Caenorhabditis elegans* homolog UNC-13 have elucidated the functions of individual domains and their binding partner in SV release. In Munc13-1, the most important Munc13 isoform in the mouse hippocampal synapses, the N-terminal C2A domain binds to the active zone protein RIM, releasing Munc13-1 from an autoinhibitory homodimerization and promoting SV priming and exocytosis (*Camacho et al., 2017*; *Deng et al., 2011*). The C1 and the C2B domains, binding to diacylglycerol (DAG) and phosphatidylinositol 4,5-bisphosphate (PIP$_2$) on the plasma membrane, are implicated in facilitating SV release and boosting synaptic potentiation (*Basu et al., 2007*; *Michelassi et al., 2017*; *Shin et al., 2010*). The C-terminal region, including the C2C domain, was found to interact with the SV membrane to trigger SV exocytosis (*Padmanarayana et al., 2021*). Thus, the biochemical characteristics and the functional analysis of the domains in Munc13 support a bridge model in which Munc13 connects SVs and the plasma membrane to trigger membrane fusion and neurotransmitter release. Indeed, recent cryo-EM crystal structure analysis of the Munc13 core sequence has provided direct evidence supporting this model (*Grushin et al., 2022*; *Xu et al., 2017*). This arrangement of Munc13 is believed to expose the central MUN domain to bind to the SNARE complexes and initiate the SNARE assembly (*Lai et al., 2017*; *Ma et al., 2011*; *Ma et al., 2013*; *Xu et al., 2017*; *Yang et al., 2015*). Given the functional diversity of the multiple domains in Munc13, their coordination and integration likely encompass key mechanisms of synaptic transmission and synaptic plasticity.

The cryo-EM crystal structure (*Grushin et al., 2022*) has highlighted the crucial role of C1 and C2B membrane interaction in supporting the Munc13 bridge and facilitating SV exocytosis. However, our previous studies showed that deletion of either C1 or C2B in *C. elegans* UNC-13 significantly enhanced SV release (*Li et al., 2019*; *Liu et al., 2021*; *Michelassi et al., 2017*). The results were mimicked by the HK mutation (histidine to lysine) in C1 that disrupts the DAG binding of C1 and by the Ca$^{2+}$-binding mutations in C2B (aspartate to asparagine, also referred to as DN mutations) which disrupt Ca$^{2+}$-dependent PIP$_2$ binding of C2B (*Basu et al., 2007*; *Li et al., 2019*; *Michelassi et al.,*

*2017*). It has been postulated that the HK mutation in Munc13-1 (H567K) lowers the energy barrier of SV exocytosis, thereby enhancing the probability of neurotransmitter release (*Basu et al., 2007*). Thus, evidence from both invertebrates and vertebrates supports the notion that disrupting the membrane interaction of C1 and C2B turns Munc13/UNC-13 into a gain-of-function state. Autoinhibition models have been proposed to explain the enhanced SV release induced by C1 and C2B disruption (*Basu et al., 2007*; *Michelassi et al., 2017*). According to this model, under basal conditions, C1 and C2B adopt autoinhibitory conformations which prevent them from binding to the plasma membrane. The elevated levels of DAG and $PIP_2$ on the membrane alleviate the autoinhibition of C1 and C2B, enabling them to bind the membrane and enhance SV release.

Despite the previous findings, the mechanisms underlying the inhibitory roles of C1 and C2B as well as their disinhibition to enhance synaptic transmission remain unclear, as does the modulation of their functions by other domains within Munc13/UNC-13. Here, we showed that the HK and DN mutations in C1 and C2B converted UNC-13 into a gain-of-function state, leading to increased SV release. Whereas the concurrent mutations of HK and DN eliminated the increase of SV release, reverting UNC-13 to its basal physiological state. This indicates that C1 and C2B are functionally dependent and supports a mutual inhibition model in which C1 and C2B inhibit each other under basal conditions. Disrupting the membrane interaction of either domain releases the other, allowing it to bind to the membrane to enhance release. This model was supported by the fact that individual C1 or C2B domains remarkably enhanced synaptic transmission. In addition, we found that the N-terminal C2A and X domains in UNC-13 differentially regulate C1 and C2B functions. The effects of HK and DN mutations were significantly strengthened by the absence of C2A but were eliminated by the absence of X, indicating distinct roles for C2A and X during the functional switch of UNC-13 between basal physiological state and gain-of-function state. Furthermore, we found that the polybasic motif of C2B plays a critical role in SV release. Finally, we showed that the domain coordination in the short UNC-13 isoform controls the protein switch between the gain-of-function state and the loss-of-function state. Together, our results revealed novel coordinated mechanisms by which the major domains in UNC-13 regulate SV release, providing significant insight into synaptic transmission and plasticity.

## Results

### Disrupting the membrane interaction of C1 and C2B turns UNC-13 into gain-of-function state

Previous biochemical and structural studies have established C1 and C2B as the primary domains in Munc13 that mediate interaction with the plasma membrane (*Kazanietz et al., 1995*; *Michelassi et al., 2017*; *Shin et al., 2010*). In *C. elegans*, there are two UNC-13 isoforms, a long isoform called UNC-13L and a short isoform known as UNC-13S (also called UNC-13MR) (*Hu et al., 2013*). Both C1 and C2B in *C. elegans* UNC-13, rat Munc13, and fly UNC-13 are highly conserved, including the residues that mediate their DAG and $PIP_2$ binding (*Figure 1A*). To delineate the contribution of their membrane interaction to SV release, we introduced the HK and DN mutations into the *C. elegans* UNC-13. The UNC-13L constructs carrying the HK or DN mutations were expressed in the nervous system (under the *snb-1* promoter) of the *unc-13* null mutants (*s69*) to assess the rescue of SV release. The *C. elegans* neuromuscular junction (NMJ) consists of cholinergic and GABAergic synapses which release acetylcholine and GABA as excitatory and inhibitory neurotransmitters, respectively (*Vashlishan et al., 2008*). Measurement of SV release at the NMJ synapses was conducted through electrophysiological recordings of miniature excitatory and inhibitory postsynaptic currents (mEPSCs and mIPSCs, commonly referred to as minis or spontaneous release) and electric stimulus-evoked EPSCs. Note that evoked EPSCs were recorded under 1 mM $Ca^{2+}$ condition, while mEPSCs and mIPSCs were recorded under both 0 mM and 1 mM $Ca^{2+}$ conditions. Our previous studies have indicated that certain *C. elegans* mutants display more pronounced defects in spontaneous release under conditions of either absence of $Ca^{2+}$ or low $Ca^{2+}$ conditions.

Compared to the severe effects of the HK mutation in Munc13-1 (*Basu et al., 2007*; *Lou et al., 2008*), the HK mutation in the worm UNC-13L (H696K, referred to as $L^{HK}$; *Figure 1A*) produced a modest increase in SV release, predominantly observed in the GABAergic synapses in the absence of extracellular $Ca^{2+}$. The mIPSC frequency in animals rescued with $L^{HK}$ exhibited a significant 50% increase in 0 mM $Ca^{2+}$ condition (*Figure 1G and H*), while evoked EPSCs and mEPSCs remained

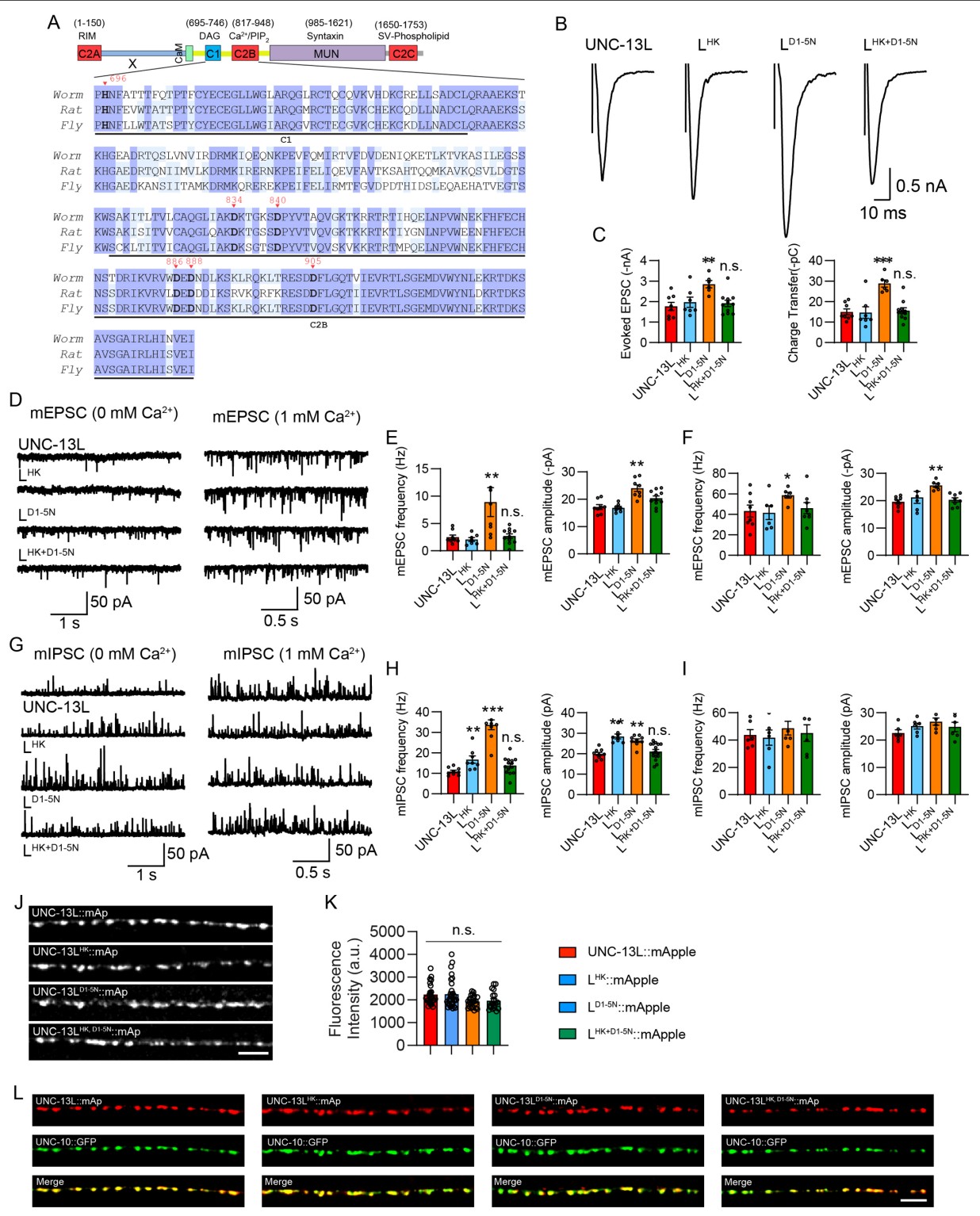

**Figure 1.** Concurrent HK and DN mutations eliminate the enhancement of synaptic vesicle (SV) release. (**A**) Sequence alignment of the C1 and C2B domains between the worm UNC-13L, the rat Munc13-1, and the fly UNC-13A. The highly conserved residues in all three species are marked in blue, while shared only by two species are marked in light blue. The HK mutation in C1 that affects diacylglycerol (DAG) binding and the DN mutations in C2B that abolish $Ca^{2+}$ binding are marked bold and indicated by inverted red triangles. (**B**) Example traces of stimulus-evoked excitatory postsynaptic currents (EPSCs) from UNC-13L, $L^{HK}$, $L^{D1-5N}$, and $L^{HK+D1-5N}$ rescued animals. (**C**) Quantification of the evoked EPSC amplitude and charge transfer from the same genotypes as in B. (**D**) Representative miniature EPSC (mEPSC) traces (recorded at 0 mM and 1 mM $Ca^{2+}$) from the indicated genotypes. (**E and**

*Figure 1 continued on next page*

*Figure 1 continued*

F) Averaged mEPSC frequency and amplitude from the same genotypes as in D. (**G**) Representative miniature inhibitory postsynaptic current (mIPSC) traces (recorded at 0 mM and 1 mM Ca$^{2+}$) from the indicated genotypes. (**H and I**) Quantification of the frequency and amplitude of the mIPSCs from the same genotypes as in G. (**J**) Representative confocal z stack images for mApple-tagged UNC-13L, UNC-13L$^{HK}$, UNC-13L$^{DN}$, and UNC-13L$^{HK+DN}$ (all driven by the *unc-129* promoter). Scale bar, 5 µm. (**K**) Quantification of the fluorescence intensity from the lines in J. (**L**) Colocalization between the active zone marker UNC-10::GFP and mApple-tagged UNC-13L, UNC-13L$^{HK}$, UNC-13L$^{DN}$, and UNC-13L$^{HK+DN}$. Data are mean ± SEM (**p<0.01, ***p<0.001 when compared to UNC-13L rescue; n.s., nonsignificant when compared to UNC-13L rescue; one-way ANOVA).

The online version of this article includes the following figure supplement(s) for figure 1:

**Figure supplement 1.** Concurrent HK and D3,4N mutations also eliminate the enhancement of synaptic vesicle (SV) release.

unchanged (*Figure 1B and C*), suggesting a relatively weak impact of the HK mutation on worm UNC-13L function.

Like the C2B domains in other synaptic proteins such as Synaptotagmin-1, the C2B domain in Munc13 and UNC-13 also contains five conserved aspartates that coordinate two Ca$^{2+}$ ions binding to C2B and facilitate Ca$^{2+}$-dependent phospholipid binding (*Shin et al., 2010*). Prior studies have demonstrated that mutating the first and second aspartates in Munc13-1 C2B (D1,2N) or the third and fourth aspartates in UNC-13L C2B (D3,4N) leads to an increase in the probability of neurotransmitter release (*Michelassi et al., 2017*; *Shin et al., 2010*). To maximize the effect of the DN mutations on SV release, we mutated all five aspartates in UNC-13L C2B (D834N, D840N, D886N, D888N, and D905N, termed L$^{D1-5N}$; *Figure 1A*). We found that the animals rescued with L$^{D1-5N}$ exhibited remarkable increases in SV release, including evoked EPSCs, mEPSCs, and mIPSCs (*Figure 1B–I*). Moreover, the enhancement of SV release induced by L$^{D1-5N}$ was greater than that in the previously reported L$^{D3,4N}$ (*Supplementary file 1*). Neither the HK nor the DN mutations enhanced spontaneous release in the 1 mM Ca$^{2+}$ condition, indicating saturated effects at this Ca$^{2+}$ level. Thus, our results corroborate earlier findings in Munc13-1 and demonstrate the functional conservation of the C1 and C2B domains across species and synaptic types. Collectively, our findings demonstrate that disrupting the membrane interaction of C1 and C2B turns UNC-13 into the gain-of-function state which triggers neurotransmitter release at an elevated level.

## Concurrent HK and DN mutations eliminate the enhancement of SV release in UNC-13L

The increased SV release in animals expressing L$^{HK}$ and L$^{D1-5N}$ raised the question about the mechanism underlying the gain-of-function of UNC-13. To address this question, we performed a screen in UNC-13L to examine the functional involvement of other major domains in the L$^{HK}$ and the L$^{D1-5N}$ context. Remarkably, we found that the effects of the HK and DN mutations were entirely abolished by the concurrent mutations in UNC-13L (L$^{HK+D1-5N}$). SV release in animals rescued with L$^{HK+D1-5N}$ is akin to those in the UNC-13L rescued animals, with no significant differences observed (*Figure 1B–I*). Our results thus revealed the intriguing dependence of the HK-induced enhancement in SV release on C2B binding Ca$^{2+}$ and PIP$_2$, while the DN-induced enhancement in SV release requires the interaction of C1 and DAG.

To determine whether the changes of SV release by mutated UNC-13L are attributed to the expression level of UNC-13 proteins, we generated fusion proteins of wild-type and mutated UNC-13L tagged with mApple (UNC-13L::mApple, L$^{HK}$::mApple, L$^{D1-5N}$::mApple, and L$^{HK+D1-5N}$::mApple, all were driven by the *unc-129* promoter) and examined their abundance in the nervous system. Our results showed that the HK, DN, and HK+DN mutations did not alter the expression level of UNC-13, with the fluorescence intensity being indistinguishable from all strains (*Figure 1J and K*). These results indicate that SV release changes induced by HK and DN mutations resulted from functional defects of C1 and C2B. The localization of UNC-13 proteins at synapses was also examined by analyzing the colocalization with the active zone marker UNC-10::GFP (*unc-10* encodes the mammalian RIM). Our results showed that all UNC-13 proteins, including wild-type and mutated UNC-13L (all tagged with mApple), exhibited a well colocalization with UNC-10 (*Figure 1L*), demonstrating that the HK and DN mutations do not alter the synaptic localization of UNC-13.

We further confirmed the above observation by examining SV release in UNC-13L carrying concurrent mutations of HK and D3,4N. We found that although the D3,4N mutations in C2B resulted in relatively weaker changes compared to D1-5N (*Supplementary file 1*), the HK and D3,4N concurrent

mutations decreased SV release to the UNC-13L level, similar to the results in HK and D1-5N (*Figure 1— figure supplement 1*). Thus, our findings support the notion of functional interdependence between C1 and C2B domains in boosting SV release.

The above results indicate that disrupting the membrane interaction of C1 and C2B simultaneously restores UNC-13 to its basal physiological state, implying that under the basal conditions, neither C1 nor C2B can effectively interact with the plasma membrane. Prior studies have proposed that C1 inhibits the MUN domain activity by an intramolecular interaction with the Munc13 C terminus (*Basu et al., 2007*), and that binding of C1 to DAG relieves this inhibition on the MUN domain. This relief of inhibition likely arises from a conformational change in C1 following DAG binding. However, our results provide an alternative interpretation, suggesting that C1 and C2B mutually inhibit each other in the basal state. The HK or DN mutations induce conformational changes in either C1 or C2B, thereby releasing the inhibition on the other domain, allowing it to bind to the membrane and trigger release. This mutual inhibition model also explains why simultaneous HK and DN mutations eliminate the enhancement of SV release.

## The N-terminus regulates the functions of C1 and C2B

The above results support the notion that the HK and DN mutations induce a functional switch of UNC-13 between basal physiological and gain-of-function states. We next sought to investigate whether alternations in the N-terminal sequence of UNC-13 contribute to changes in its functional status. The N-terminus of UNC-13L contains a C2A domain and an X domain (an unstructured region with a calmodulin-binding site, by AlphaFold; *Figure 1A*), both of which have been demonstrated to play pivotal roles in SV release (*Li et al., 2019*; *Liu et al., 2019*). The HK and DN mutations were introduced into UNC-13R, a truncated UNC-13 lacking the whole N-terminal sequence of UNC-13L (1–607 aa). We found that compared to the relatively modest changes in SV release in $L^{HK}$, the HK mutation in UNC-13R ($R^{HK}$) led to significantly greater enhancements in both spontaneous release (0 mM $Ca^{2+}$, mIPSC frequency, $R^{HK}/R=1.9$ vs $L^{HK}/L=1.5$) and evoked release (charge transfer, $R^{HK}/R=1.7$, $L^{HK}/L=0.95$; *Figure 2B–I*). Conversely, the DN mutations in UNC-13R ($R^{D1-5N}$) produced a comparable increase in evoked EPSCs relative to $L^{D1-5N}$, but caused a weaker enhancement in spontaneous release (0 mM $Ca^{2+}$, mEPSC frequency $R^{D1-5N}/R=2.1$ vs $L^{D1-5N}/L=3.6$, mIPSC frequency $R^{D1-5N}/R=1.5$ vs $L^{D1-5N}/L=3.1$, *Figure 2B–I*). These results suggest that the N-terminus of UNC-13L does indeed influence the functions of C1 and C2B.

Despite the differential impacts on the effects of HK and DN mutations, we found that the N terminus in UNC-13L did not alter the functional interdependence of C1 and C2B. Our results showed that the HK- and DN-induced enhancement in SV release was completely suppressed by the concurrent mutations of HK and DN in UNC-13R ($R^{HK+D1-5N}$). Electrophysiological recordings of evoked EPSCs, mEPSCs, and mIPSCs in $R^{HK+D1-5N}$ rescued animals were indistinguishable from those in UNC-13R animals (*Figure 2B–I*), indicating that these concurrent mutations restore UNC-13R to the basal physiological state, consistent with the observations in UNC-13L. Given that in UNC-13R, C1 and C2B represent the only two domains situated in the N-terminal region of the central MUN domain, the results obtained in $R^{HK+D1-5N}$ animals further validate the mutual inhibition model between C1 and C2B. These findings, coupled with those observed in UNC-13L, demonstrate that UNC-13 undergoes a functional switch between its basal physiological state and its gain-of-function state through changes in the membrane interactions of C1 and C2B.

## C2A and X play different roles in regulating the functions of C1 and C2B

The distinct effects of the HK and DN mutations in UNC-13R suggest that the C2A and X domains within the N-terminus of UNC-13L may exert differing influences on the functions of C1 and C2B. Our previous studies have revealed that the C2A domain and the X domain in UNC-13L exhibit contrasting regulatory roles in SV release. Specifically, deletion of either the C2A or X domain in UNC-13L results in a reduction or augmentation in both spontaneous and evoked release (*Liu et al., 2019*). To further explore the potential impacts of C2A and X on C1 and C2B function, we introduced the HK, DN, and HK+DN mutations into UNC-13ΔC2A (ΔC2A$^{HK}$, ΔC2A$^{D1-5N}$, ΔC2A$^{HK+D1-5N}$) and UNC-13ΔX (ΔX$^{HK}$, ΔX$^{D1-5N}$, ΔX$^{HK+D1-5N}$) and examined their effects on SV release.

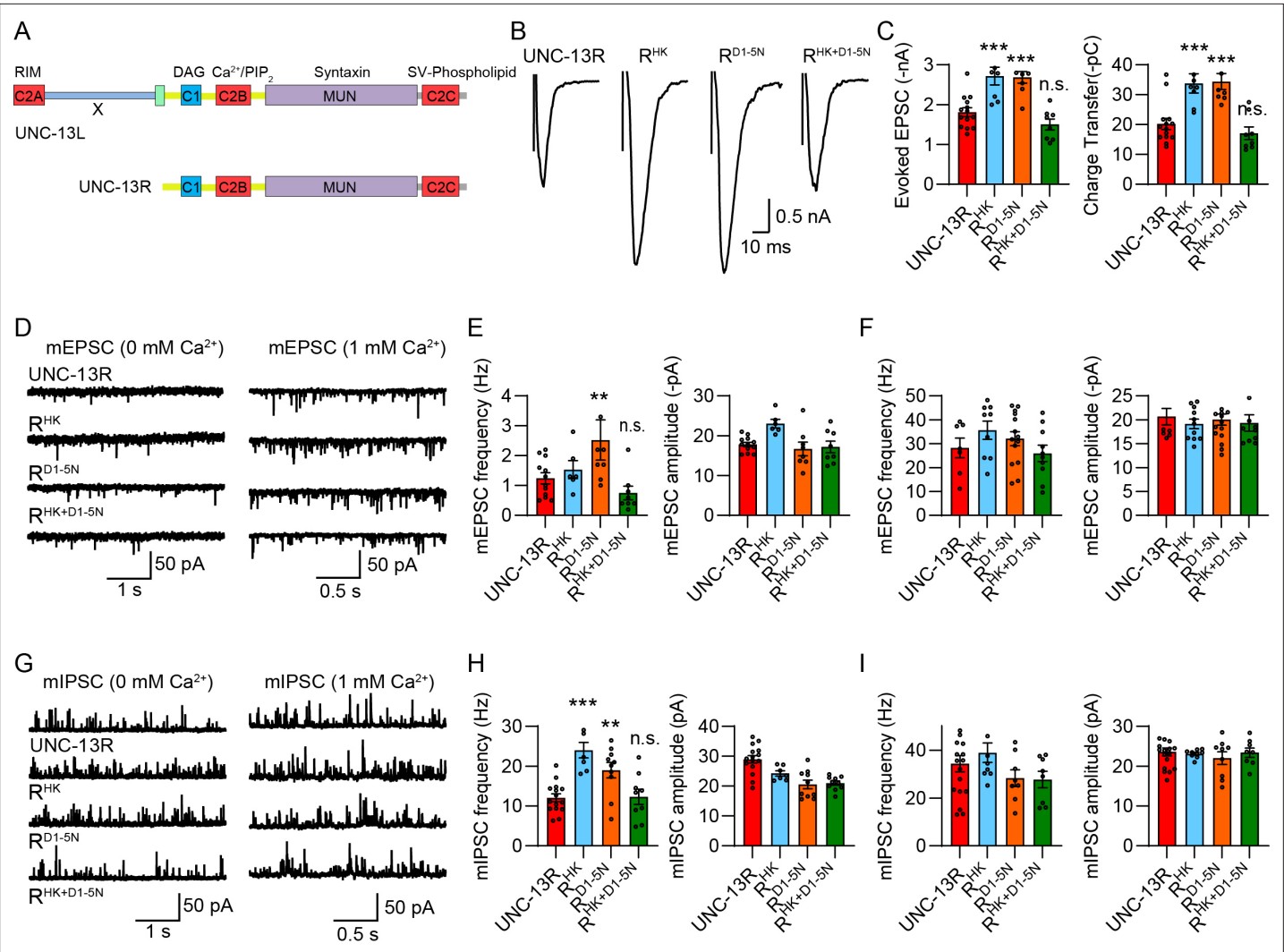

**Figure 2.** The N-terminus in UNC-13L regulates the functions of C1 and C2B. (**A**) Cartoon depicting the HK and DN mutations in UNC-13L and UNC-13R. (**B**) Example traces of stimulus-evoked excitatory postsynaptic currents (EPSCs) from UNC-13R, $R^{HK}$, $R^{D1-5N}$, and $R^{HK+D1-5N}$ rescued animals. (**C**) Quantification of the evoked EPSC amplitude and charge transfer from the same genotypes as in B. The dashed lines represent the level of wild-type UNC-13L rescue. (**D**) Representative miniature EPSC (mEPSC) traces (recorded at 0 mM and 1 mM $Ca^{2+}$) from the indicated genotypes. (**E and F**) Quantification of the frequency and amplitude of the mEPSCs from the same genotypes as in D. (**G**) Representative miniature inhibitory postsynaptic current (mIPSC) traces (recorded at 0 mM and 1 mM $Ca^{2+}$) from the indicated genotypes. (**H and I**) Quantification of the frequency and amplitude of the mIPSCs from the same genotypes as in G. Data are mean ± SEM (**p<0.01, ***p<0.001 when compared to UNC-13L rescue; n.s., nonsignificant when compared to UNC-13L rescue; one-way ANOVA).

Compared to UNC-13L, UNC-13ΔC2A exhibited a partial rescue of SV release in *unc-13* mutants, with the frequencies of mEPSCs and mIPSCs, as well as the amplitude and charge transfer of evoked EPSC being significantly lower than those in UNC-13L rescued animals (***Supplementary file 1***). However, the HK mutation in UNC-13ΔC2A led to a markedly higher increase in SV release relative to $L^{HK}$, including enhancements in evoked EPSCs and mIPSCs (***Figure 3B–I***). Similarly, the DN mutations in UNC-13ΔC2A also produced a stronger increase in evoked EPSCs (***Figure 3B–C***) and a comparable increase in mEPSCs and mIPSCs compared to $L^{D1-5N}$ (***Figure 3D–I***). These results thus support the notion that the C2A domain negatively regulates the transition of UNC-13 from its basal physiological state to its gain-of-function state, likely by reinforcing the mutual inhibition between C1 and C2B. Moreover, the concurrent HK and DN mutations in UNC-13ΔC2A did not result in a decrease in SV release compared to $ΔC2A^{HK}$ and $ΔC2A^{D1-5N}$, indicating that $ΔC2A^{HK+D1-5N}$ still acts in a gain-of-function state relative to UNC-13ΔC2A. Thus, our results indicate that the C2A domain not only impedes the

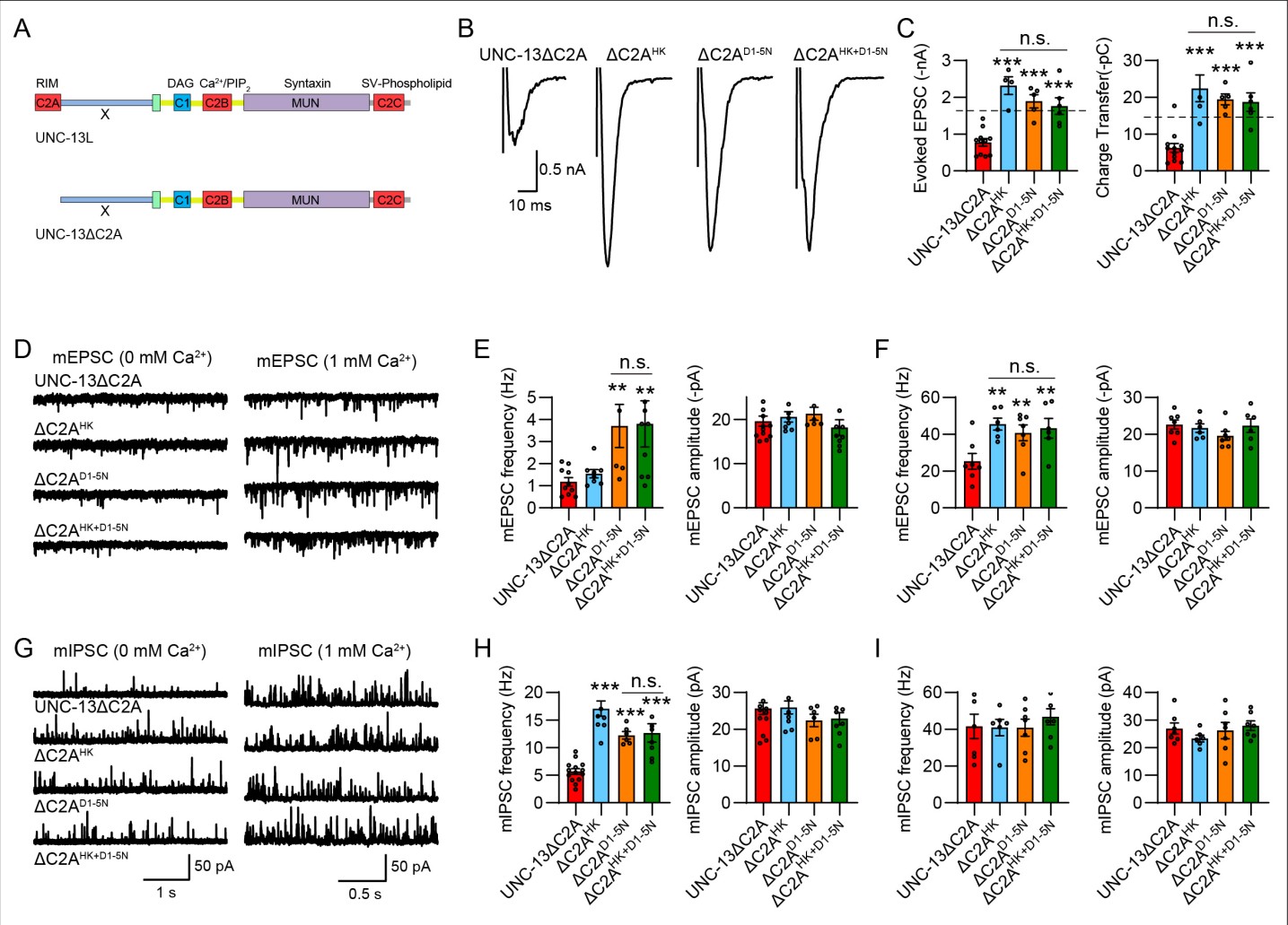

**Figure 3.** The C2A domain regulates the functions of C1 and C2B. (**A**) Domain structure of UNC-13L and UNC-13LΔC2A. (**B**) Example traces of stimulus-evoked excitatory postsynaptic currents (EPSCs) from UNC-13ΔC2A, ΔC2A$^{HK}$, ΔC2A$^{D1-5N}$, and ΔC2A$^{HK+D1-5N}$ rescued animals. (**C**) Quantification of the evoked EPSC amplitude and charge transfer from the same genotypes as in B. The dashed lines represent the level of wild-type UNC-13L rescue. (**D**) Representative miniature EPSC (mEPSC) traces (recorded at 0 mM and 1 mM Ca$^{2+}$) from the indicated genotypes. (**E and F**) Quantification of the frequency and amplitude of the mEPSCs from the same genotypes as in (**D**). (**G**) Representative miniature inhibitory postsynaptic current (mIPSC) traces (recorded at 0 mM and 1 mM Ca$^{2+}$) from the indicated genotypes. (**H and I**) Quantification of the frequency and amplitude of the mIPSCs from the same genotypes as in G. Data are mean ± SEM (**p<0.01, ***p<0.001 when compared to UNC-13ΔC2A rescue; n.s., nonsignificant; one-way ANOVA).

conversion of UNC-13 from the basal physiological state to the gain-of-function state, but also facilitates the reversion of UNC-13 from its gain-of-function state to its basal physiological state.

To determine the impact of the X domain on the functions of C1 and C2B, we examined SV release in *unc-13* mutants rescued with UNC-13ΔX, ΔX$^{HK}$, ΔX$^{D1-5N}$, and ΔX$^{HK+D1-5N}$. Consistent with our previous findings (***Li et al., 2019***), the deletion of the X domain in UNC-13L resulted in a notable increase in SV release (***Supplementary file 1***). Unexpectedly, neither the HK nor the DN mutations induced changes in evoked neurotransmitter release. The animals rescued with ΔX$^{HK}$, ΔX$^{D1-5N}$, and ΔX$^{HK+D1-5N}$ exhibited evoked EPSCs comparable to those in the UNC-13ΔX animals (***Figure 4B and C***), suggesting that the X domain facilitates the switch of UNC-13 to the gain-of-function state, likely by alleviating the mutual inhibition between C1 and C2B.

Although the HK and DN mutations did not impact evoked release in UNC-13ΔX, these mutations still elicited significant increases in spontaneous release. Compared to UNC-13ΔX, both ΔX$^{HK}$ and ΔX$^{D1-5N}$ significantly enhanced mEPSC and mIPSC frequencies (***Figure 4D–I***). The discrepant effects of ΔX$^{HK}$ and ΔX$^{D1-5N}$ in evoked and spontaneous release likely arise from the distinct regulatory mechanisms governing these two major forms of neurotransmitter release (***Liu et al., 2018***). Moreover, the heightened mEPSCs

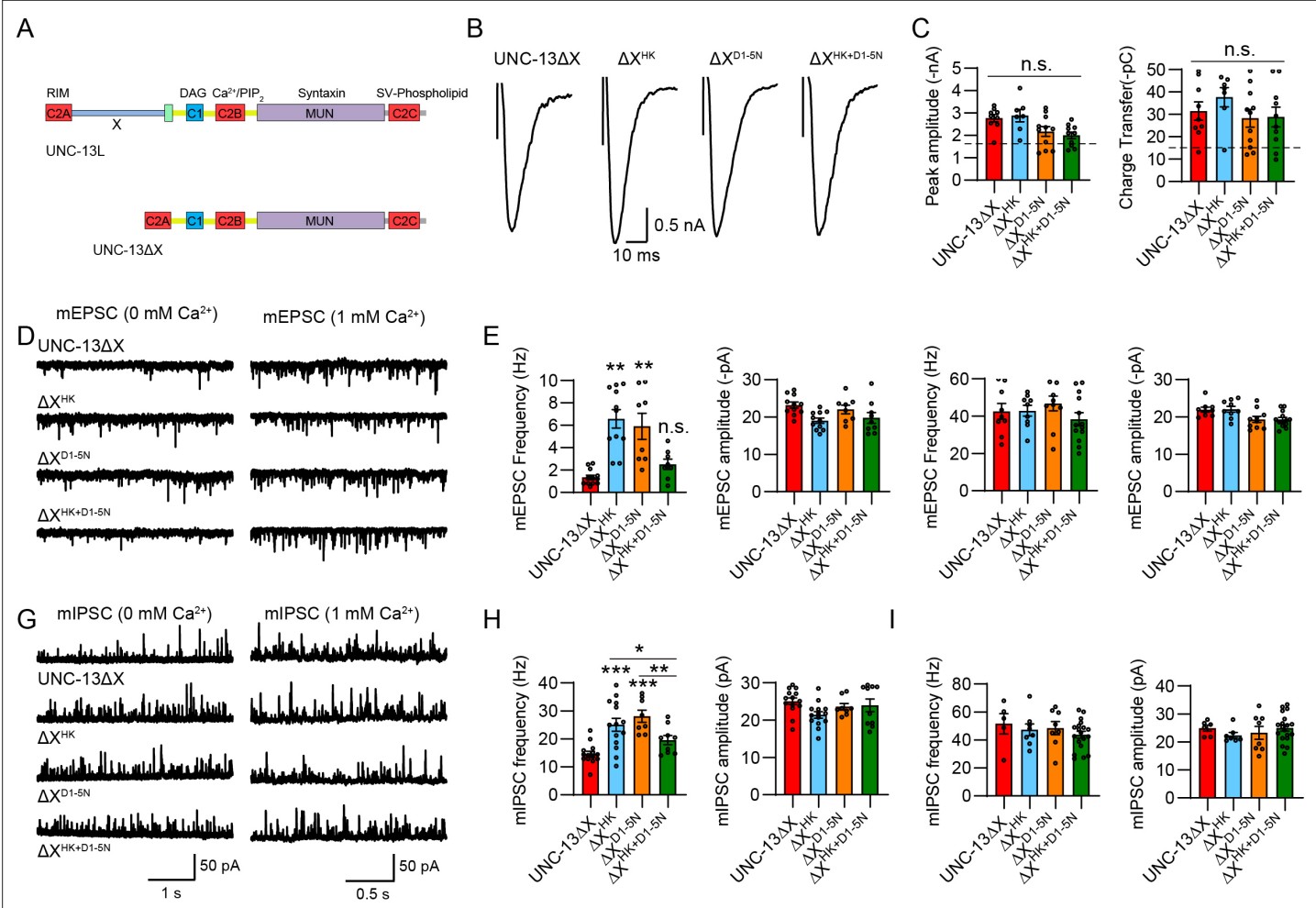

**Figure 4.** The X domain regulates the functions of C1 and C2B. (**A**) Domain structure of UNC-13L and UNC-13ΔX. (**B**) Example traces of stimulus-evoked excitatory postsynaptic currents (EPSCs) from UNC-13ΔX, ΔX^HK, ΔX^D1-5N, and ΔX^HK+D1-5N rescued animals. (**C**) Quantification of the evoked EPSC amplitude and charge transfer from the same genotypes as in B. The dashed lines represent the level of wild-type UNC-13L rescue. Data are mean ± SEM (n.s., nonsignificant; one-way ANOVA). (**D**) Representative miniature EPSC (mEPSC) traces (recorded at 0 mM and 1 mM Ca²⁺) from the indicated genotypes. (**E and F**) Quantification of the frequency and amplitude of the mEPSCs from the same genotypes as in D. Data are mean ± SEM (**p<0.01 when compared to UNC-13ΔX rescue; n.s., nonsignificant when compared to UNC-13ΔX rescue; one-way ANOVA). (**G**) Representative miniature inhibitory postsynaptic current (mIPSC) traces (recorded at 0 mM and 1 mM Ca²⁺) from the indicated genotypes. (**H and I**) Quantification of the frequency and amplitude of the mIPSCs from the same genotypes as in G. Data are mean ± SEM (*p<0.05, **p<0.01, ***p<0.001 when compared to UNC-13ΔX rescue; one-way ANOVA).

in ΔX^HK and ΔX^D1-5N animals were completely suppressed in ΔX^HK+D1-5N animals (*Figure 4E*), whereas the mIPSCs were partially reduced (*Figure 4H*). These findings suggest that in the absence of the X domain, C1 and C2B still exhibit functional interdependence in regulating spontaneous release.

Taken together, our results demonstrate that the N-terminal C2A and X domains exert differential impacts on C1 and C2B functions, particularly in the regulation of evoked neurotransmitter release. The C2A domain appears to inhibit the switch of UNC-13 into a gain-of-function state induced by C1 and C2B, while promoting its switch back to the basal physiological state. In contrast, the X domain facilitates the switch of UNC-13 into a gain-of-function state induced by C1 and C2B. Thus, our findings revealed novel mechanisms through which C1 and C2B, in coordination with C2A and X, govern the functional switch of UNC-13 between basal physiological and gain-of-function states, thereby modulating synaptic transmission and synaptic plasticity.

## Individual C1 and C2B promotes SV release in a gain-of-function manner

The above findings have demonstrated that the gain-of-function of UNC-13 hinges on the membrane interaction of C1 and C2B. Specifically, C1 binding to DAG is essential for enhancing SV release when C2B's membrane interaction is disrupted by the DN mutations, whereas C2B binding to $Ca^{2+}$ and $PIP_2$ is crucial for enhancing SV release when C1's membrane interaction is blocked by the HK mutation. These results align with previous observations indicating that C1 and C2B act in concert to control SV release (*Michelassi et al., 2017*). Notably, disrupting the membrane interaction of one domain within the C1-C2B module reveals the intrinsic function of the other domain in triggering SV release, as evidenced by the decreased SV release observed when the HK mutation is introduced into the $R^{D1-5N}$ background and vice versa (*Figure 2*).

Building upon these observations, we proposed a hypothesis that individual C1 or C2B domains may directly enhance SV release by interacting with the plasma membrane. To test this hypothesis, we isolated the individual C1 and C2B domains in UNC-13 and fused them to MUNC2C, the minimal fragment known to trigger SV release (termed as C1MUNC2C and C2BMUNC2C, *Figure 5A*; *Basu et al., 2005*; *Madison et al., 2005*; *Stevens et al., 2005*). We found that while MUNC2C only partially rescued evoked EPSCs compared to UNC-13L, C1MUNC2C robustly triggered evoked EPSCs to a higher level than UNC-13L (*Figure 5B and C*, *Supplementary file 1*). The increased evoked EPSCs in C1MUNC2C were completely suppressed by the HK mutation (C1$^{HK}$MUNC2C), confirming the role of C1 in promoting SV release by binding to DAG. Moreover, the evoked charge transfer in C1MUNC2C is significantly higher than that in UNC-13R (26.3 pc vs 18.5 pc) which contains both C1 and C2B, suggesting that the C1 domain in C1MUNC2C is not inhibited by C2B and acts in a gain-of-function state.

Similarly, the C2B domain also strongly enhanced evoked EPSCs when fused to MUNC2C, doubling the evoked EPSCs compared to UNC-13R (charge transfer 36 pc vs 18.5 pc; *Figure 5B and C*, *Supplementary file 1*). However, the DN mutations in C2BMUNC2C (C2B$^{D1-5N}$MUNC2C) did not diminish the evoked EPSCs, suggesting additional functions of C2B. One possibility is that the linker region between C2B and MUN (termed linker3; *Figure 5A*) is necessary for C2B to exert its function properly. Indeed, adding the linker3 in chimeric protein C2BMUNC2C (C2Blinker3MUNC2C) did not further increase SV release but enabled the proteins to be responsible for the DN mutations. The animals rescued with C2B$^{D1-5N}$linker3MUNC2C exhibited significantly smaller evoked EPSCs compared to C2Blinker3MUNC2C animals (*Figure 5B and C*), demonstrating the necessity of linker3 for C2B function, likely by enhancing its flexibility and enabling conformational changes upon binding to $Ca^{2+}$ and $PIP_2$ to trigger SV release. Thus, like the observations in C1MUNC2C, our results indicate that in C2Blinker3MUNC2C, the C2B domain acts in a gain-of-function mode because of the lack of inhibition from C1.

Compared to MUNC2C, the C1MUNC2C also notably enhanced mEPSC and mIPSC frequencies, with such enhancements being suppressed by the HK mutation (*Figure 5D–G*), mirroring observations in evoked release. Interestingly, although C2B dramatically enhanced spontaneous release, the DN mutations in C2BMUNC2C and C2Blinker3MUNC2C did not cause a decrease in mEPSC and mIPSC frequencies. These findings suggest that the C2B domain possesses additional functions that stimulate the spontaneous release and part of the evoked release, as indicated by the larger remaining evoked EPSCs in C2B$^{D1-5N}$linker3MUNC2C animals compared to MUNC2C animals (*Figure 5B and C*).

## Mutual inhibition between C1 and C2B requires the linker regions

In UNC-13L and UNC-13R, the arrangement of C1, C2B, and MUN is mediated by several linkers (referred to as linker1, linker2, and linker3; *Figure 2A*). To elucidate how C1 and C2B establish mutual inhibition, we generated a chimeric protein, C1C2BMUNC2C, devoid of the linkers. If C1 and C2B contribute to exert inhibitory effects on each other, we would anticipate a reduction in SV release comparable to MUNC2C levels in animals expressing C1C2BMUNC2C. However, our findings showed that the evoked EPSCs mediated by C1C2BMUNC2C were similar to those mediated by C1MUNC2C and C2BMUNC2C, and are significantly higher than those in UNC-13R (*Figure 5—figure supplement 1A*, *Figure 5B and C*). Nevertheless, the spontaneous release was notably diminished in C1C2BMUNC2C animals compared to C1MUNC2C and C2BMUNC2C, albeit partially (*Figure 5—figure supplement 1B–E*). These results suggest that the mutual inhibition of C1 and C2B relies on

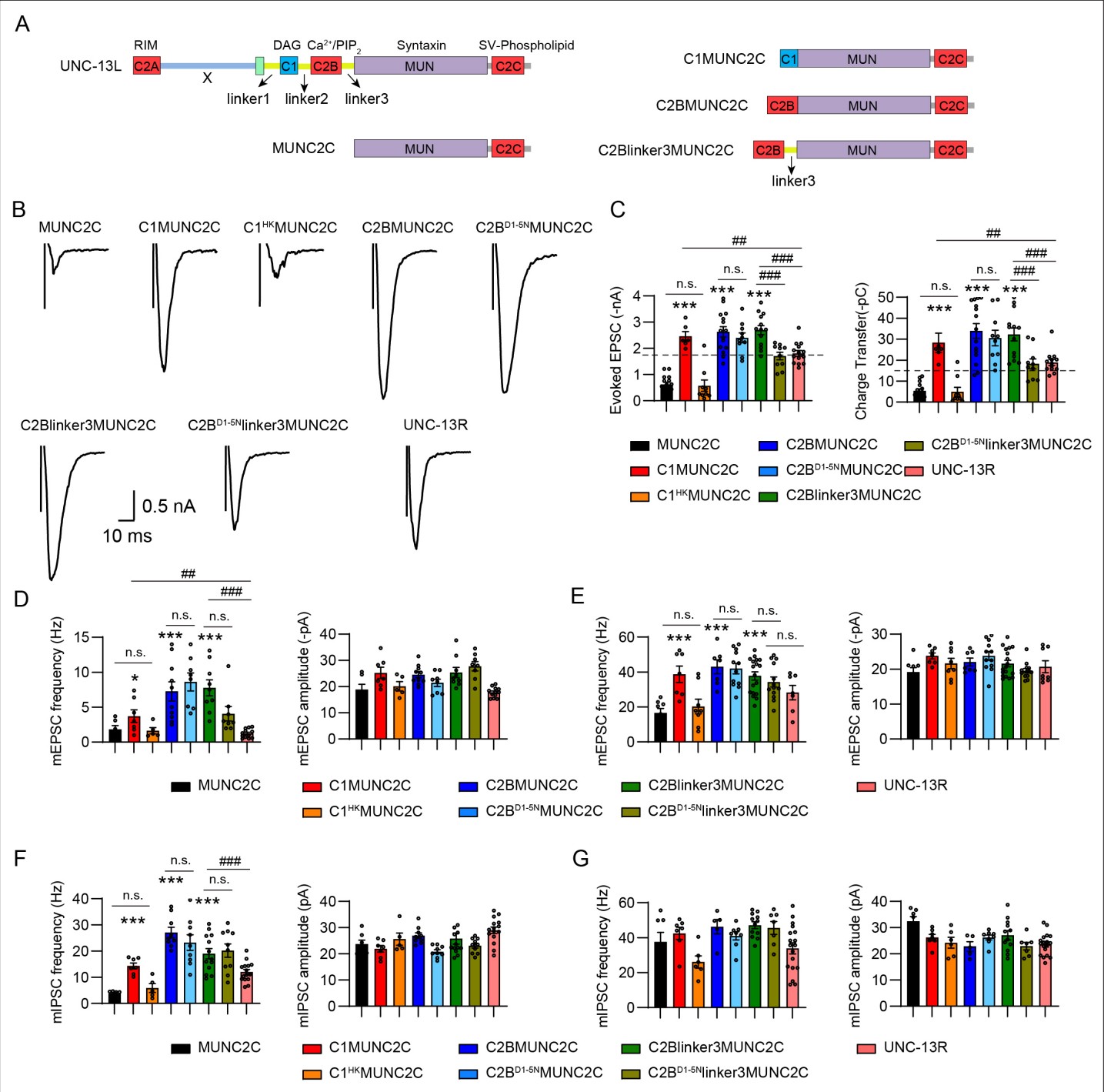

**Figure 5.** Individual C1 and C2B promotes synaptic vesicle (SV) release in a gain-of-function manner. (**A**) Cartoons depicting UNC-13L, MUNC2C, C1MUNC2C, C2BMUNC2C, and C2Blinker3MUNC2C. (**B**) Example traces of stimulus-evoked excitatory postsynaptic currents (EPSCs) from the indicated genotypes. (**C**) Quantification of the evoked EPSC amplitude and charge transfer from the same genotypes as in B. The dashed lines represent the level of wild-type UNC-13L rescue. (**D and E**) Averaged miniature EPSC (mEPSC) frequency and amplitude (recorded at 0 mM and 1 mM Ca$^{2+}$) from the same genotypes as in B. (**F and G**) Quantification of the miniature inhibitory postsynaptic current (mIPSC) frequency and amplitude (recorded at 0 mM and 1 mM Ca$^{2+}$) from the same genotypes as in B. Data are mean ± SEM (*p<0.05, ***p<0.001 when compared to MUNC2C rescue; ##p<0.01, ###p<0.001; n.s., nonsignificant; one-way ANOVA).

The online version of this article includes the following figure supplement(s) for figure 5:

**Figure supplement 1.** The mutual inhibition between C1 and C2B requires the linkers.

the presence of the linkers, presumably due to their role in stabilizing C1 and C2B in an inhibitory state.

## The polybasic motif is critical for the C2B function

To explore the additional functions of the C2B domain, we focused on the polybasic motif that commonly exists in C2B-containing synaptic proteins. Prior studies have reported the critical role of the polybasic motif in mediating the function of the C2B domain (*Fernandez et al., 2001*; *Kuo et al., 2009*; *Sato et al., 2010*; *Tsuboi et al., 2007*; *Wu et al., 2022*). Residues within the polybasic motif carry positive charges, enabling them to potentially engage in a $Ca^{2+}$-independent electrostatic interaction with the plasma membrane (*Kuo et al., 2009*; *Wu et al., 2022*). Mutations affecting these polybasic residues have been linked to synaptic transmission defects (*Wu et al., 2022*). A sequence alignment between the C2B domains of the worm UNC-13 and rat Munc13-1 (*Figure 6A*), as well as two other synaptic proteins synaptotagmin-1 and rabphilin-3A (*Figure 6—figure supplement 1*), revealed the presence of the polybasic motif situated in the third or fourth β sheet according to the C2B structure (K849, K851, R852, R853, and R855 in UNC-13L; *Figure 6A*). This observation suggests that the C2B domain may also facilitate SV release through interactions mediated by the polybasic motif with the plasma membrane.

We next examined the functional significance of the polybasic motif in C2B by neutralizing all five residues in C2BMUNC2C and C2Blinker3MUNC2C (KKRRR/QQQQQ, denoted as K/Q in this study; *Figure 6A*). Our results showed that the polybasic mutations in C2Blinker3MUNC2C (C2B$^{K/Q}$linker3MUNC2C) nearly abolished spontaneous and evoked neurotransmitter release. The mEPSCs, mIPSCs, and evoked EPSCs in C2B$^{D1-5N}$linker3MUNC2C animals were all reduced to levels similar to those in MUNC2C (*Figure 6B–I*), indicating the critical role of the polybasic motif in C2B domain function. Furthermore, the polybasic mutations in C2BMUNC2C also resulted in a severe reduction in SV release, except for the mEPSCs in 1 mM $Ca^{2+}$ (*Figure 6F*), suggesting that the regulation of C2B on SV release through the polybasic motif is independent of the linker3. Unexpectedly, the mIPSC frequency in 1 mM $Ca^{2+}$ was decreased to an even lower level than that in MUNC2C animals by the polybasic mutations (*Figure 6I*). These results might be attributed to the polybasic mutations affecting the function of MUNC2C. Taken together, our results demonstrate that the C2B domain in UNC-13 operates through at least two mechanisms to regulate SV release: by binding to $PIP_2$ through the $Ca^{2+}$-binding loops and by interacting with the plasma membrane via the polybasic motif.

Surprisingly, we found that the polybasic mutations in UNC-13L did not lead to synaptic transmission defects. Neither spontaneous nor evoked EPSCs were altered in UNC-13L$^{K/Q}$ rescued animals (*Figure 6—figure supplement 2*). Several possibilities may account for these unexpected results. First, it is conceivable that the C2B domain carrying polybasic mutations may still function in UNC-13L through its ability to bind $Ca^{2+}$ and $PIP_2$. However, this scenario seems unlikely given that the two operational modes of C2B did not demonstrate functional redundancy in C2Blinker3MUNC2C (*Figures 5 and 6*). Second, the effects of the polybasic mutations in UNC-13L might be compensated by other domains such as C1. The compensation mechanism could maintain SV release at a normal level but not a higher level as observed in the DN mutations. We observed pronounced effects of the polybasic mutations in C2Blinker3MUNC2C, suggesting the absence of such a compensation mechanism.

## Disrupting C1 and C2B membrane interaction simultaneously inactivates UNC-13S

In addition to UNC-13L, the short isoform UNC-13S also plays a pivotal role in *C. elegans* neurons regulating synaptic transmission. Unlike UNC-13L, which harbors C2A or X in the N-terminus, UNC-13S features an N-terminal M domain (also known as UNC-13MR; *Figure 7A*). Our previous studies have revealed distinct localization patterns of UNC-13L and UNC-13S at the active zones of worm NMJ synapses, each triggers SV exocytosis with unique synaptic properties (*Hu et al., 2013*). Furthermore, deletion of either C1 or C2B in UNC-13S has been shown to augment SV release (*Liu et al., 2021*), suggesting similar roles of C1 and C2B in both UNC-13 isoforms. Moreover, our studies have demonstrated that the M domain exerts an inhibitory effect on UNC-13S function by modulating the activity of C1 and C2B. Prompted by these prior findings, we tested the idea that C1 and C2B also mutually inhibit each other in UNC-13S and the potential involvement of the M domain.

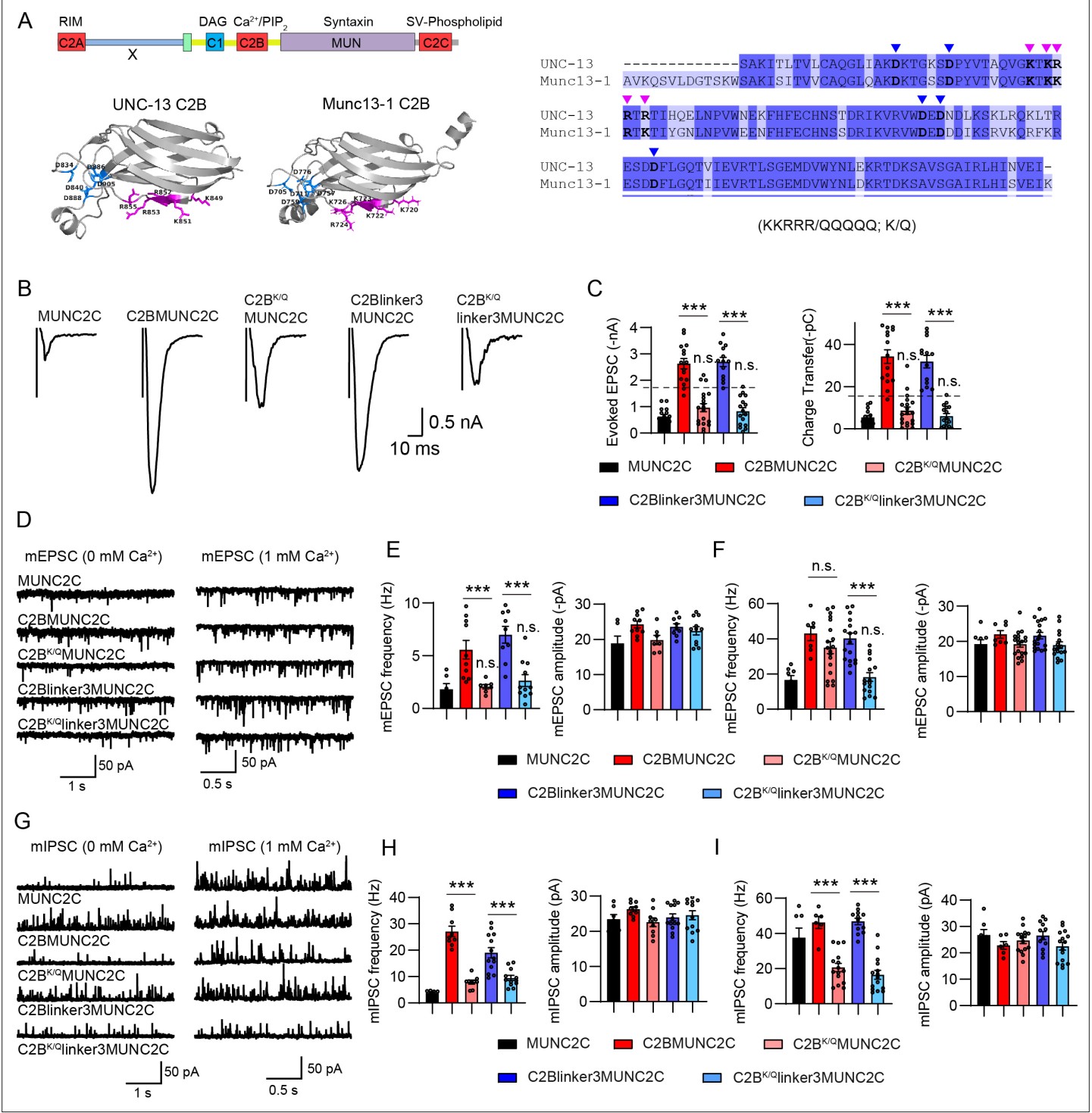

**Figure 6.** The polybasic motif is critical for the C2B function. (**A**) Left, crystal structures of UNC-13 C2B (EBI AlphaFold database, UniProt ID: P27715) and Munc13-1 C2B (from PDB: 7T7V). The polybasic residues in each C2B are labeled in purple, and the five $Ca^{2+}$-binding sites (aspartates) are labeled in blue. Right, sequence comparison of UNC-13 C2B and Munc13-1 C2B. The polybasic residues are indicated by purple arrowheads, and the five aspartates are indicated by blue arrowheads. (**B**) Example traces of stimulus-evoked excitatory postsynaptic currents (EPSCs) from MUNC2C, C2BMUNC2C, C2B^KQMUNC2C, C2Blinker3MUNC2C, and C2B^KQlinker3MUNC2C rescued animals. (**C**) Quantification of the evoked EPSC amplitude and charge transfer from the same genotypes as in B. The dashed lines represent the level of wild-type UNC-13L rescue. (**D**) Representative miniature EPSC (mEPSC) traces (recorded at 0 mM and 1 mM $Ca^{2+}$) from the indicated genotypes. The dashed lines represent the level of wild-type UNC-13L rescue. (**E and F**) Quantification of the frequency and amplitude of the mEPSCs. (**G**) Representative miniature inhibitory postsynaptic current (mIPSC) traces

*Figure 6 continued on next page*

*Figure 6 continued*

(recorded at 0 mM and 1 mM Ca$^{2+}$) from the indicated genotypes in B. (**H and I**) Quantification of the frequency and amplitude of the mIPSCs. Data are mean ± SEM (\*\*\*p<0.001; n.s., nonsignificant when compared to MUNC2C; one-way ANOVA).

The online version of this article includes the following figure supplement(s) for figure 6:

**Figure supplement 1.** The polybasic motifs in synaptotagmin-1 and rabphilin-3A.

**Figure supplement 2.** The polybasic mutations in UNC-13L do not affect synaptic vesicle (SV) release.

By introducing the HK, DN, and concurrent HK and DN mutations into UNC-13S (termed S$^{HK}$, S$^{D1-5N}$, S$^{HK+D1-5N}$), we analyzed their effects in SV release. Consistent with our previous findings, UNC-13S can only partially restore SV release (*Supplementary file 1*), with the release kinetics of the evoked EPSCs being slower than that in the wild-type (*Figure 7—figure supplement 1A*). Despite this, the HK and the DN mutations in UNC-13S notably enhanced evoked EPSCs to a level similar to that in UNC-13R (*Figure 7B and C*). The effects of HK and DN on evoked EPSCs appear to be independent of the speed of SV release, as the evoked EPSCs mediated by UNC-13S and UNC-13R have slower risetime but were enhanced to a similarly higher level by the HK and DN mutations (*Figure 7—figure supplement 1A and B*). Although mEPSCs remained unchanged, mIPSCs were significantly increased by HK and DN mutations in UNC-13S, surpassing the enhancement observed in UNC-13R. These results suggest that C1 and C2B perform similar fundamental functions across different UNC-13 isoforms, with the N-terminal M domain seemingly modulating C1 and C2B functions.

Surprisingly, the concurrent HK and DN mutations in UNC-13S nearly abolished SV release. In S$^{HK+D1-5N}$ animals, the evoked EPSCs, mEPSCs, and mIPSCs were all dramatically decreased compared to UNC-13S animals (*Figure 7B–I*). These results diverge from observations in UNC-13L and UNC-13R, where the concurrent mutations merely reduced SV release to the control level (*Figures 1 and 2*). This suggests that disrupting the membrane interaction of C1 and C2B simultaneously renders UNC-13S inactive, transitioning it from a gain-of-function state to a loss-of-function state. The striking reduction in SV release in S$^{HK+D1-5N}$ animals as well as the increase in SV release in S$^{HK}$ and S$^{D1-5N}$ animals did not arise from changes in expression level of UNC-13S (*Figure 7J and K*). It appeared that the N-terminal M domain in UNC-13S plays a pivotal role in orchestrating this transition. Given the dominant inhibitory effect of the M domain on SV release in UNC-13S (*Liu et al., 2021*), the inactivation induced by the concurrent mutations likely arises from a reinforced inhibition by the M domain. In summary, our results revealed the presence of mutual inhibition between C1 and C2B in UNC-13S, with the M domain predominantly promoting the conversion of UNC-13S from the gain-of-function state to the loss-of-function state.

## Mutual inhibition is a specific characteristic of C1 and C2B

To determine whether the observed mutual inhibition in UNC-13 is C1 and C2B specific, we introduced a PL mutation in the linker3 region between C2B and MUN (*Figure 8*). The PL mutation, identified in the human UNC13A, has been associated with a dyskinetic movement disorder, developmental delay, and autism (*Lipstein et al., 2017*). In mouse Mucn13-1, the PL mutation dramatically enhances SV release in cultured hippocampus neurons (*Lipstein et al., 2017*). *C. elegans* mutants carrying the PL mutation display hypersensitivity to acetylcholinesterase inhibitor aldicarb (*Lipstein et al., 2017*), suggesting increased SV release. Given that linker3 is adjacent to C2B, we tested whether the concurrent mutations of HK and PL could produce similar effects as the HK and DN mutations in UNC-13L and UNC-13S, respectively.

Our results revealed that the PL mutation in UNC-13L (P956L, L$^{PL}$, *Figure 8A*) did not alter evoked EPSCs (*Figure 8B and C*) but significantly enhanced mEPSCs and mIPSCs (*Figure 8F, G, L, and M*). In particular, the mIPSC frequency in 0 mM Ca$^{2+}$ was nearly doubled compared to that in UNC-13L, demonstrating a strong effect of the PL mutation on spontaneous release in UNC-13L, consistent with the observations in Munc13-1 (*Lipstein et al., 2017*). However, the concurrent HK and PL mutations did not cause a decrease in mEPSC and mIPSC frequencies, unlike the concurrent HK and DN mutations in UNC-13L and UNC-13R, which decreased SV release to control levels (*Figures 1 and 2*). In UNC-13S, the PL mutation (P608L, S$^{PL}$, *Figure 8A*) moderately increased evoked EPSCs but remarkably enhanced them when combined with the HK mutation (*Figure 8D and E*), contrasting with the nearly abolished evoked EPSCs observed with HK and DN mutations (*Figure 7B and C*). Similarly,

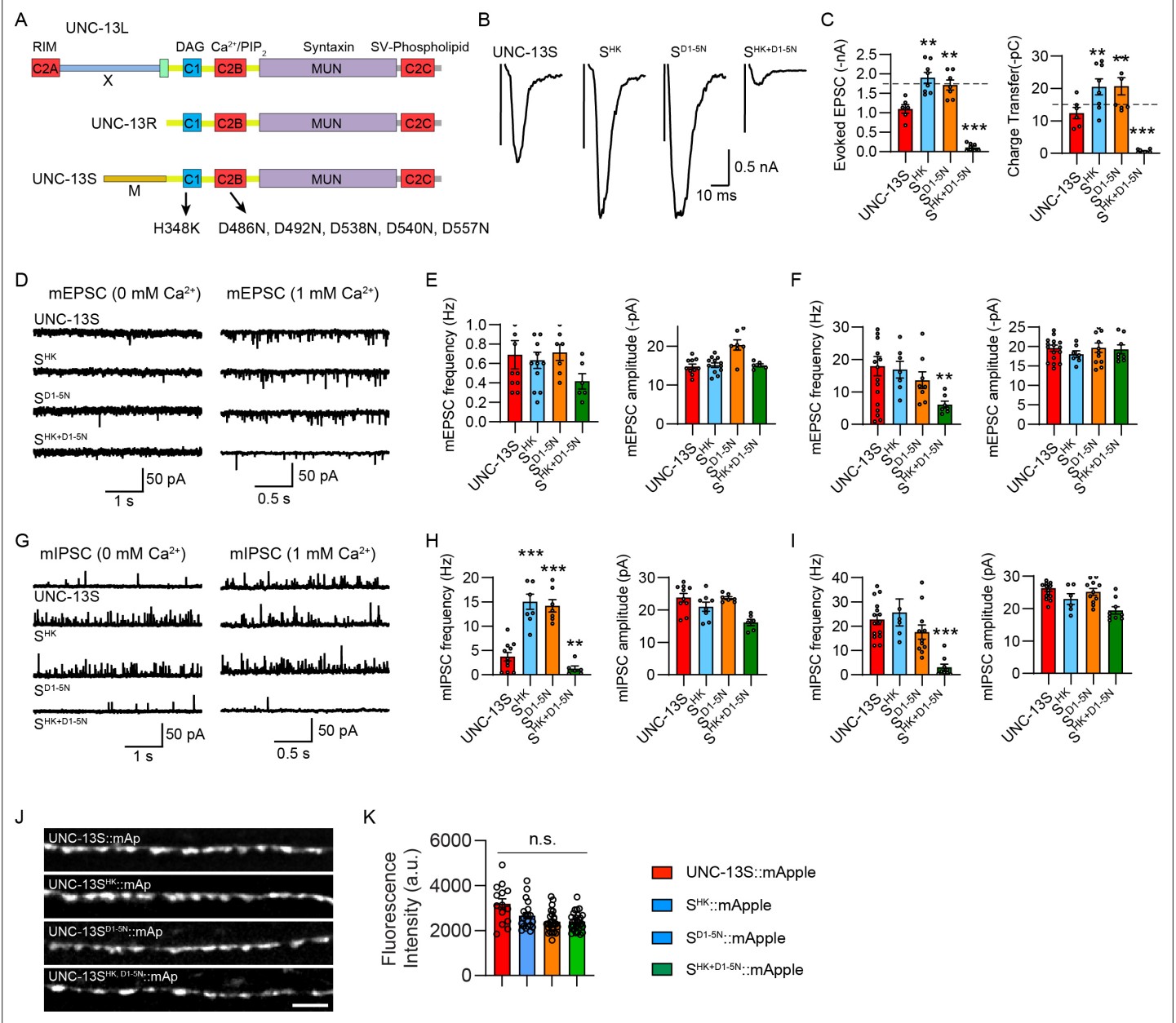

**Figure 7.** Disrupting C1 and C2B membrane interaction simultaneously results in the inactivation of UNC-13S. (**A**) Cartoon depicting the HK and DN mutations in UNC-13L, UNC-13R, and UNC-13S. (**B**) Example traces of stimulus-evoked excitatory postsynaptic currents (EPSCs) from UNC-13S, $S^{HK}$, $S^{D1-5N}$, $S^{HK+D1-5N}$ rescued animals. (**C**) Averaged evoked EPSC amplitude and charge transfer. The dashed lines represent the level of wild-type UNC-13L rescue. (**D**) Representative miniature EPSC (mEPSC) traces (recorded at 0 mM and 1 mM $Ca^{2+}$) from the indicated genotypes in B. (**E and F**) Quantification of the frequency and amplitude of the mEPSCs. (**G**) Representative miniature inhibitory postsynaptic current (mIPSC) traces (recorded at 0 mM and 1 mM $Ca^{2+}$) from the indicated genotypes in B. (**H and I**) Quantification of the frequency and amplitude of the mIPSCs. (**J**) Representative confocal z stack images for mApple-tagged UNC-13S, $S^{HK}$, $S^{DN}$, and $S^{HK+DN}$ (all driven by the *unc-129* promoter). Scale bar, 5 μm. (**K**) Quantification of the fluorescence intensity. Data are mean ± SEM (**p<0.01, ***p<0.001 when compared to UNC-13S rescue; n.s., nonsignificant; one-way ANOVA).

The online version of this article includes the following figure supplement(s) for figure 7:

**Figure supplement 1.** The HK and DN mutations do not alter release kinetics in UNC-13S and UNC-13R.

spontaneous release was elevated in $S^{HK+PL}$ animals compared to $S^{HK}$ and $S^{PL}$. These findings suggest that while the PL mutation in linker3 enhanced SV release, C1 and linker3 do not exhibit functional coordination and mutual inhibition as observed with C1 and C2B. Thus, our results further confirm that mutual inhibition in UNC-13 is a specific characteristic of the C1-C2B module.

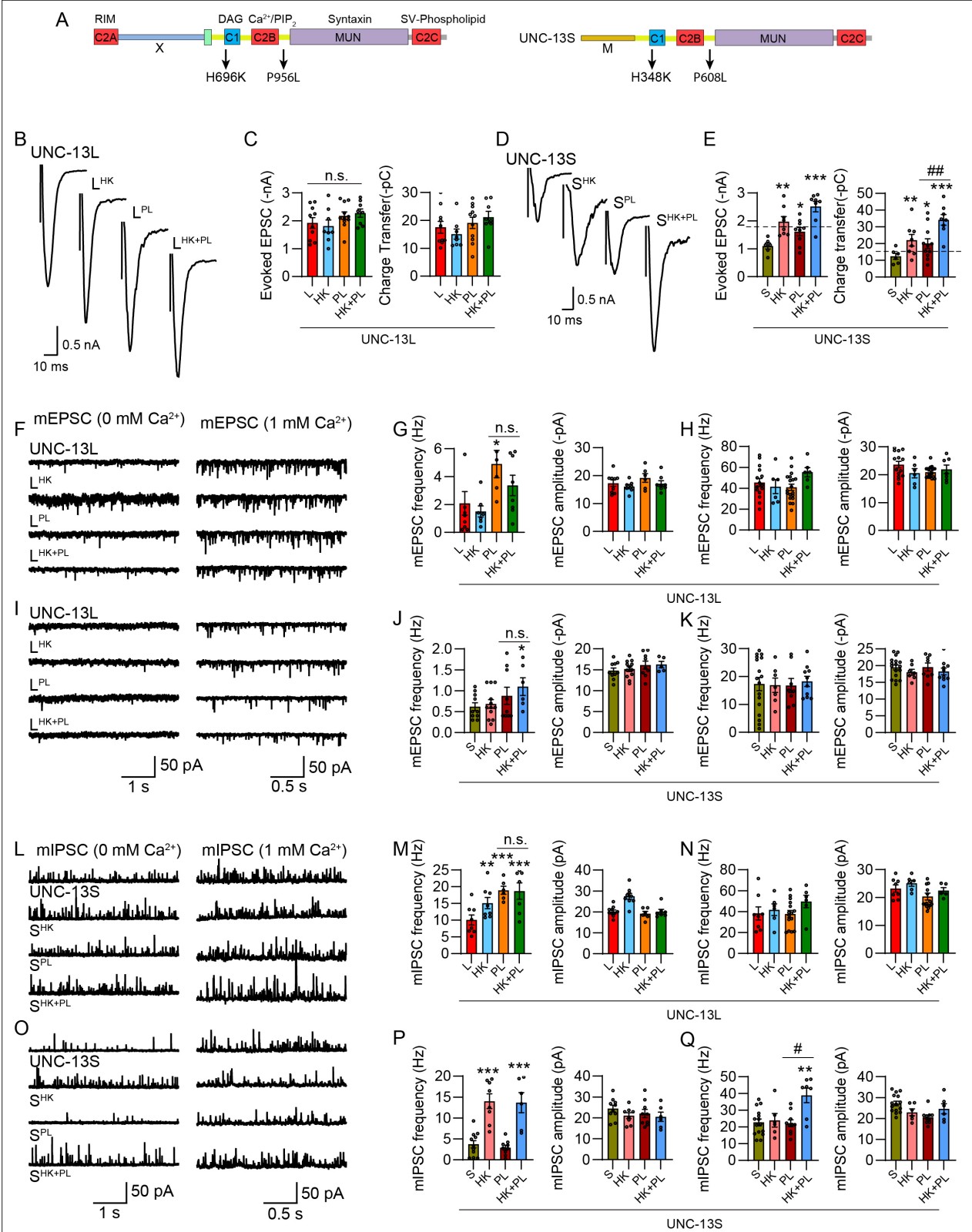

**Figure 8.** The concurrent mutations of HK and PL in UNC-13L and UNC-13S do not decrease synaptic vesicle (SV) release. (**A**) Positions of the PL mutation in UNC-13L and UNC-13S. (**B, D**) Example traces of stimulus-evoked excitatory postsynaptic currents (EPSCs) from *unc-13* mutants rescued with wild-type UNC-13L and S, and UNC-13L or UNC-13S carrying HK, PL, and HK+PL mutations. (**C, E**) Quantification of the evoked EPSC amplitude and charge transfer from the same genotypes as in B and D. The dashed lines represent the level of wild-type UNC-13L rescue. (**F, I**) Representative

*Figure 8 continued on next page*

*Figure 8 continued*

miniature EPSC (mEPSC) traces (recorded at 0 mM and 1 mM $Ca^{2+}$) from the same genotypes in B and D. (**G–K**) Quantification of the frequency and amplitude of the mEPSCs (G and J, 0 mM $Ca^{2+}$; H and K, 1 mM $Ca^{2+}$). (**L, O**) Representative miniature inhibitory postsynaptic current (mIPSC) traces (recorded at 0 mM and 1 mM $Ca^{2+}$) from the indicated genotypes. (**M–Q**) Averaged mIPSC frequency and amplitude in 0 mM $Ca^{2+}$ (**M, P**) and 0 mM $Ca^{2+}$ (**N, Q**). Data are mean ± SEM (*p<0.05, **p<0.01, ***p<0.001 when compared to UNC-13L or UNC-13S rescue; #p<0.05, ##p<0.01; n.s., nonsignificant; one-way ANOVA).

## Discussion

Our study presents several significant findings on the mechanisms underlying synaptic transmission regulated by the key synaptic regulator UNC-13. These findings contribute novel insights into the intricate coordination of UNC-13's multiple functional domains and their impact on synaptic function. First, we elucidated a mutual inhibition mechanism between the C1 and C2B domains of UNC-13, highlighting their pivotal roles in the regulation of basal synaptic transmission under physiological conditions. Disrupting the membrane interaction of either domain led to a gain-of-function state, enhancing SV release. Whereas the concurrent mutations in both domains abolished the enhancement of SV release, restoring UNC-13 to the basal physiological state. Second, we validated the mutual inhibition model by isolating individual C1 and C2B domains, demonstrating that each domain alone promotes SV release at a significantly higher level than the presence of both domains, indicating that either C1 or C2B acts in the gain-of-function state without the inhibition from the other. Third, we revealed the involvement of the N-terminal C2A and X domains in the functional switch of UNC-13 induced by disruption of C1 and C2B membrane interaction. Specifically, the C2A domain inhibits UNC-13 from converting to the gain-of-function state and also promotes its restoration to the basal physiological state. The X domain, however, promotes the transition of UNC-13 to the gain-of-function state. Fourth, we uncovered the multifaceted role of the C2B domain in triggering SV release, including its interaction with $Ca^{2+}/PIP_2$ and its regulation through the polybasic motif, which likely mediates the plasma membrane interaction. Fifth, we demonstrated how domain coordination in the short UNC-13 isoform controls the protein switch between gain-of-function and loss-of-function states, offering insights into isoform-specific regulatory mechanisms. Overall, these findings revealed previously unknown mechanisms underlying UNC-13-mediated synaptic transmission regulation, and provided significant contributions to our understanding of synaptic transmission and short-term synaptic plasticity.

### Mutual inhibition between C1 and C2B in the basal physiological state

Previous studies in both vertebrates and invertebrates have established the inhibitory roles of C1 and C2B within mammalian Munc13 and worm UNC-13 (*Basu et al., 2007*; *Michelassi et al., 2017*). Notably, while treatment with phorbol ester, an analog of DAG, remarkably enhances SV release, the HK mutation, which disrupts C1 binding to DAG and phorbol ester, also leads to a comparable increase in SV release (*Basu et al., 2007*; *Lou et al., 2008*). This suggests that the HK mutation mimics C1 activation. A proposed disinhibition model posits that the C1 domain, when bound to the Munc13 C terminus, inhibits the catalytic MUN domain. However, binding of C1 to DAG or phorbol ester leads to a conformational change of C1, thereby disinhibiting the MUN domain and enhancing release. Recent studies in the worm UNC-13 showed that deleting C2B or mutating the $Ca^{2+}$-binding sites in loop 3 of C2B also enhanced SV release, revealing the inhibitory function of the C2B domain in UNC-13. Moreover, C1 and C2B appear to act in concert inhibiting SV release, with the elevation in DAG and $Ca^{2+}$ relieving this inhibition, thereby enhancing SV release (*Michelassi et al., 2017*).

Our findings contribute significantly to the existing autoinhibition models of C1 and C2B and support a mutual inhibition model between C1 and C2B, thereby advancing our understanding of how membrane binding mutations lead to a gain-of-function state of UNC-13. Through a screen investigating the functional involvement of other domains in the UNC-13L gain-of-function, we discovered interesting observations. Specifically, concurrent HK and DN mutations in C1 and C2B abolished the enhancement in SV release, providing evidence that C1 and C2B mutually inhibit each other. Moreover, SV release levels in $L^{HK+D1-5N}$ and $R^{HK+D1-5N}$ animals were restored to those in UNC-13L and UNC-13R, indicating that C1 and C2B may not contribute to basal synaptic transmission. Instead, they form an inhibitory unit that controls baseline transmission at an acceptable level, which could optimize synaptic function by stabilizing the size of the primed vesicle pool.

The recent cryo-EM structure of C1C2BMUNC2C has provided insights into the functional importance of C1 and C2B, revealing that these two domains pack at the N-terminal end of the MUN and interact with the membrane, supporting the Munc13 bridge. While it has been widely believed that the membrane interactions of C1 and C2B are crucial for promoting SV release, direct evidence supporting their facilitatory roles has been lacking, mainly due to the mutations that disrupt C1 and C2B membrane interaction or C1 and C2B deletion increase SV release by enhancing the probability of neurotransmitter release. Our findings with C1MUNC2C and C2Blinker3MUNC2C provide novel evidence of the facilitatory roles of C1 and C2B in promoting SV release. Additionally, the HK and DN mutations in these chimeric proteins disrupted the domain function and decreased SV release, contrasting the observations in UNC-13L, UNC-13R, and UNC-13S. Notably, our results suggest that individual C1 and C2B domains within the chimeric proteins act in the gain-of-function state, as evidenced by higher SV release levels compared to UNC-13R. The decreased SV release caused by the HK and DN mutations in these chimeric proteins demonstrates that C1 and C2B promote SV release through their direct interaction with DAG and $Ca^{2+}$/$PIP_2$. Thus, our findings provide valuable insights into the long-lasting question of how C1 and C2B in Munc13/UNC-13 trigger SV release.

How do C1 and C2B inhibit each other under the basal conditions? Structure analysis has shown that the C1C2BMUNC2C fragment of Munc13 exhibits a rigid arc-like rod shape with limited flexibility (*Xu et al., 2017*). In this configuration, the DAG-binding region of the C1 domain and the $Ca^{2+}$-binding region of the C2B domain are positioned near each other with the same orientation. This arrangement is expected to facilitate cooperation between C1 and C2B in their membrane binding. Moreover, cryo-EM studies have reported two conformations of C1C2BMUNC2C, open and closed, differing mainly in the orientation of the MUN domain (*Grushin et al., 2022*). In the closed conformation, the MUN domain is titled, and both C1 and C2B interact with the membrane, allowing SVs to be positioned close to the membrane in a primed, ready-to-release state. Since the cyro-EM experiments were conducted without $Ca^{2+}$ and DAG (*Grushin et al., 2022*), it is possible that C1 and C2B can naturally form membrane-attached orientation in the SV supply chain and reach a stabilized state which represents the basal physiological state. Although direct physical interaction between C1 and C2B is not identified in structural analysis, extensive interdomain contacts, including those involving C1 and C2B, as well as C2B and MUN, have been indicated in Munc13 (*Xu et al., 2017*). These findings may support the structural basis for the mutual inhibition between C1 and C2B. We propose that the interdomain interfaces and the membrane contacts of C1 and C2B allow the formation of a relatively rigid C1-C2B module. This rigid unit functions like a balanced seesaw, preventing each side from engaging in further, stronger interactions with the membrane. However, the HK and DN mutations may induce conformation changes in either C1 or C2B, disrupting the balance of the rigid C1-C2B module and allowing the other end to more deeply insert into the membrane, consequently enhancing SV release.

## The N-terminus regulation on C1 and C2B function

The functions of the N-terminal C2A domain in Munc13 and UNC-13 have been extensively investigated in previous studies (*Camacho et al., 2017*; *Deng et al., 2011*; *Dulubova et al., 2005*; *Liu et al., 2019*; *Lu et al., 2006*). Of the four Munc13 isoforms in mammals, only two, Munc13-1 and ubMunc13-2, possess a C2A domain (*Betz et al., 2001*; *Brose et al., 1995*; *Song et al., 1998*). Structural and functional analyses have unveiled two contrasting binding modes of the C2A domain: it can form a homodimer with another C2A domain, thereby autoinhibiting Munc13 activity, or it can bind to the zinc finger domain of the active zone protein RIM, forming a heterodimer that promotes SV priming and release (*Camacho et al., 2017*; *Deng et al., 2011*; *Dulubova et al., 2005*). Mutations that disrupt this heterodimerization result in defects in priming and exocytosis (*Camacho et al., 2017*). Notably, the C2A domain in *C. elegans* UNC-13L exhibits similar regulatory mechanisms in SV release, demonstrating its functional conservation between vertebrates and invertebrates. In contrast, the X domain located in the N-terminus has not been thoroughly elucidated. This region was initially considered as a large linker between C2A and C1, with the structural information being unknown. Nevertheless, our previous findings demonstrated that deleting X in UNC-13L enhances SV release (*Li et al., 2019*), suggesting an inhibitory role of this region in UNC-13L.

Despite the wealth of previous research, investigations into the N-terminus of Munc13/UNC-13 have been conducted separately from analyses of other domains, partly due to the lack of structural information of the entire protein, hindering our understanding of the spatial folding of C2A and X, as

well as their potential interactions and coordination with other domains. Nevertheless, insights from the structure of C1C2BMUNC2C, as reported by *Xu et al., 2017*, have provided some clues. For example, the helix H1 situated between the CaM domain and the C1 domain (linker1 in UNC-13L) exhibits many contacts with neighboring regions, including the linker region between C1 and C2B (linker2 in UNC-13L), as well as the MUN domain. Notably, the helix H1 is closely packed against the MUN domain, with its N-terminus directed toward the center of the elongated rod-like structure. This arrangement suggests that the C2A and the X region may adopt a spatial conformation that interacts with either the MUN domain or the linker region between C1 and C2B. Such interactions could potentially regulate the MUN domain activity or influence the function of C1 and C2B.

Interestingly, our findings revealed the significant influence of C2A and X on the effects of the HK and DN mutations, providing compelling evidence linking the N-terminus to C1 and C2B domains. Specifically, we observed markedly stronger enhancements in SV release in UNC-13ΔC2A animals with the HK and DN mutations, whereas no changes were observed in UNC-13ΔX animals. This suggests that C2A and X play inhibitory and facilitatory roles, respectively, in regulating the function of the C1-C2B module, thereby controlling the switch of UNC-13 from the basal physiological state to the gain-of-function state. Moreover, in animals rescued with $\Delta C2A^{HK+D1-5N}$, SV release was not restored to the control level, as observed in UNC-13L and UNC-13R, indicating that $\Delta C2A^{HK+D1-5N}$ remains in the gain-of-function state. This supports the additional role of C2A in promoting the switch of UCN-13 from the gain-of-function state to the basal physiological state. These findings align with the stronger enhancements of release in $\Delta C2A^{HK}$ and $\Delta C2A^{D1-5N}$ animals, suggesting that the presence of C2A favors UNC-13 in the basal physiological state. This could be potentially achieved through C2A interactions with the MUN domain, inhibiting MUN activity, or by interacting with the C1-C2B module (either the two domains or the linker2), thereby strengthening the mutual inhibition of C1-C2B by stabilizing the module. These effects of C2A might be negated by binding to RIM (*Deng et al., 2011*). As for the X domain, the only known sequence is the CaM domain which binds calmodulin and $Ca^{2+}$ (*Dimova et al., 2006*; *Junge et al., 2004*). Previous studies have demonstrated that Munc13 binding to calmodulin modulates synaptic plasticity (*Junge et al., 2004*). One possible model for the X domain function is that the CaM domain also interacts with the C1-C2B module or the linker between them, promoting C1 and C2B binding to the membrane. This interaction may facilitate the transition of UNC-13 from the basal physiological state to the gain-of-function state.

Our findings that $S^{HK+D1-5N}$ almost eliminated SV release are striking. A comparison between these results with those in $R^{HK+D1-5N}$ indicates a potent inhibitory role of the N-terminal M domain in UNC-13S when C1 and C2B membrane interactions are simultaneously disrupted. Like the X domain in UNC-13L, structural information of the M domain has not been obtained. However, our previous studies have demonstrated that the M domain in UNC-13S plays an inhibitory role, likely by locking the MUN domain in a non-favorable conformation for SNARE assembly (*Liu et al., 2021*). Notably, mammalian bMunc13-2 and Munc13-3 also contain unstructured N-terminal sequences. Prior reports have indicated that the coiled-coil motif in the N-terminal of bMunc13-2 binds to the active zone protein ELKS1, thereby regulating the synaptic localization of bMunc13-2 and priming of SVs in hippocampal synapses (*Kawabe et al., 2017*). Moreover, at *Drosophila* olfactory synapses, the UNC13B isoform exhibits a differential localization with the C2A-containing isoform UNC13A, likely through the interactions of their N-termini with active zone proteins Bruchipilot (Brp, mammalian ELKS1 ortholog) and Syd-1 (*Fulterer et al., 2018*). These results suggest that the M domain in UNC-13S may have some unknown mechanisms that regulate SV release. Similar to the C2A and X in UNC-13L, we propose that the M domain can directly interact with the MUN domain, and this interaction could strongly impede the rotation of the MUN domain, thereby obstructing SV release.

## The physiological significance of the C1-C2B model

Although the C1-C2B functional switch model is mainly based on the dysfunction of the C1-C2B module (through HK and DN mutations), it provides a potential physiological framework for understanding how the structural balance of C1-C2B module relates to the variability of synaptic transmission in the nervous system. In the CNS, synaptic transmission is highly variable, and the temporal pattern of the presynaptic activity may require dynamic switching of the fusion machinery between different functional modes, thereby triggering synaptic transmission at various levels. Our model suggests that under conditions of high neuronal activity, the C1-C2B module may transition from a

balanced to an unbalanced state (gain-of-function state), thereby enhancing synaptic transmission. The future studies will focus on additional sequences in UNC-13 (e.g. the linkers between C1 and C2B, C2B and MUN) to investigate their potential roles in regulating synaptic transmission and their broader functional significance in UNC-13 dynamics.

# Materials and methods
## Resource availability
### Lead contact
Further information and requests for resources and reagents should be directed to and will be fulfilled by the Lead Contact, Zhitao Hu (zhtiaohu@cityu.edu.hk).

### Materials availability
All new strains and plasmids created in this study will be provided on request with no restrictions.

### Experimental model and subject details
Strain maintenance and genetic manipulation were performed as previously described (*Brenner, 1974*). Animals were cultivated at room temperature on nematode growth medium agar plates seeded with OP50 bacteria. On the day before experiments, L4 larval stage animals were transferred to fresh plates seeded with OP50 bacteria for all the electrophysiological recordings. The following strains were used:

Wild-type, N2 bristol
BC168 *unc-13(s69)*
KP6893 *nuEx1515 [Psnb-1::UNC-13L]; unc-13(s69)*
ZTH9 *hztEx19 [Psnb-1::UNC-13L$^{H696K}$]; unc-13(s69)*
ZTH555 *nuEx50 [Psnb-1::UNC-13L$^{D1-5N}$]; unc-13(s69)*
ZTH1148 *hztEx214 [Psnb-1::UNC-13L$^{HK, D1-5N}$]; unc-13(s69)*
JSD0835 *tauEx313 [Psnb-1::UNC-13L$^{D3,4N}$]; unc-13(s69)*
ZTH445 *hztEx87 [Psnb-1::UNC-13L$^{HK, D3,4N}$]; unc-13(s69)*
ZTH99 *hztEx20 [Psnb-1::UNC-13R]; unc-13(s69)*
ZTH424 *hztEx85 [Psnb-1::UNC-13R$^{HK}$]; unc-13(s69)*
ZTH1155 *hztEx207 [Psnb-1::UNC-13R$^{D1-5N}$]; unc-13(s69)*
ZTH1128 *hztEx205 [Psnb-1::UNC-13R$^{HK, D1-5N}$]; unc-13(s69)*
ZTH6 *hztEx106 [Psnb-1::UNC-13L(ΔC2A)]; unc-13(s69)*
ZTH440 *hztEx117 [Psnb-1::UNC-13L(ΔC2A)$^{HK}$]; unc-13(s69)*
ZTH1211 *hztEx301 [Psnb-1::UNC-13L(ΔC2A)$^{D1-5N}$]; unc-13(s69)*
ZTH1214 *hztEx303 [Psnb-1::UNC-13L(ΔC2A$^{JHK, D1-5N}$]; unc-13(s69)*
ZTH318 *hztEx41 [Psnb-1::UNC-13L(ΔX)]; unc-13(s69)*
ZTH1254 *hztEx309 [Psnb-1::UNC-13L(ΔX)$^{HK}$]; unc-13(s69)*
ZTH1239 *hztEx306 [Psnb-1::UNC-13L(ΔX)$^{D1-5N}$]; unc-13(s69)*
ZTH1240 *hztEx308 [Psnb-1::UNC-13L(ΔX)$^{HK, D1-5N}$]; unc-13(s69)*
ZTH442 *hztEx107 [Psnb-1::UNC-13(MUNC2C)]; unc-13(s69)*
ZTH589 *hztEx122 [Psnb-1::UNC-13(C1-MUNC2C)]; unc-13(s69)*
ZTH724 *hztEx124 [Psnb-1::UNC-13(C1$^{HK}$-MUNC2C)]; unc-13(s69)*
ZTH569 *hztEx120 [Psnb-1::UNC-13(C2B-MUNC2C)]; unc-13(s69)*
ZTH1019 *hztEx198 [Psnb-1::UNC-13(C2B$^{D1-5N}$-MUNC2C)]; unc-13(s69)*
ZTH976 *hztEx126 [Psnb-1::UNC-13(C2B-linker3-MUNC2C)]; unc-13(s69)*
ZTH1008 *hztEx197 [Psnb-1::UNC-13(C2B$^{D1-5N}$-linker3-MUNC2C)]; unc-13(s69)*
ZTH1101 *hztEx210 [Psnb-1::UNC-13(C2B$^{K/Q}$-MUNC2C)]; unc-13(s69)*
ZTH1085 *hztEx203 [Psnb-1::UNC-13(C2B$^{K/Q}$-linker3-MUNC2C)]; unc-13(s69)*
ZTH1103 *hztEx204 [Psnb-1::UNC-13(C1-C2B-MUNC2C)]; unc-13(s69)*
ZTH11 *hztEx11 [Psnb-1::UNC-13S]; unc-13(s69)*
ZTH421 *hztEx54 [Psnb-1::UNC-13S$^{H348K}$]; unc-13(s69)*
ZTH458 *hztEx202 [Psnb-1::UNC-13S$^{D1-5N}$]; unc-13(s69)*

ZTH1141 *hztEx206 [Psnb-1::UNC-13S$^{HK, D1-5N}$]; unc-13(s69)*
JSD0832 *tauEx310 [Psnb-1::UNC-13L$^{P956L}$]; unc-13(s69)*
ZTH1256 *hztEx313 [Psnb-1::UNC-13S$^{P608L}$]; unc-13(s69)*
ZTH1226 *hztEx305 [Psnb-1::UNC-13L$^{HK, P956L}$]; unc-13(s69)*
ZTH1244 *hztEx311 [Psnb-1::UNC-13S$^{HK, P608L}$]; unc-13(s69)*
ZTH1354 *hztEx321 [Punc-129::UNC-13L::mApple]; NuIs165 [Punc-129::UNC-10::GFP]*
ZTH1365 *hztEx327 [Punc-129::UNC-13L$^{H696K}$::mApple]; NuIs165 [Punc-129::UNC-10::GFP]*
ZTH1363 *hztEx325 [Punc-129::UNC-13L$^{D1-5N}$::mApple]; NuIs165 [Punc-129::UNC-10::GFP]*
ZTH1350 *hztEx318 [Punc-129::UNC-13L$^{HK, D1-5N}$::mApple]; NuIs165 [Punc-129::UNC-10::GFP]*
ZTH941 *hztEx129 [Punc-129::UNC-13S::mApple]; NuIs165 [Punc-129::UNC-10::GFP]*
ZTH1353 *hztEx320 [Punc-129::UNC-13 S$^{HK}$::mApple]; NuIs165 [Punc-129::UNC-10::GFP]*
ZTH1355 *hztEx323 [Punc-129::UNC-13S$^{1-5N}$::mApple]; NuIs165 [Punc-129::UNC-10::GFP]*
ZTH1339 *hztEx316 [Punc-129::UNC-13S$^{HK, D1-5N}$::mApple]; NuIs165 [Punc-129::UNC-10::GFP]*

## Method details

### Constructs, transgenes, and germline transformation

The *snb-1* promoter (3 kb) and *unc-129* promoter (2.6 kb) were inserted into JB6 vector between the SphI and BamHI sites, all the UNC-13 expression constructs were amplified by PCR and inserted into JB6 vector between the KpnI and NotI sites. Red fluorescent protein mApple was inserted between NotI and MluI sites in-frame with unc-13 genes for imaging experiments. Transgenic strains were isolated by microinjection of various plasmids using Pmyo-2::NLS-mCherry (KP#1480) as the co-injection marker.

### Electrophysiology

Electrophysiology was conducted on dissected *C. elegans* as previously described (; *Liu et al., 2019*). Worms were superfused in an extracellular solution containing 127 mM NaCl, 5 mM KCl, 26 mM NaHCO$_3$, 1.25 mM NaH$_2$PO$_4$, 20 mM glucose, 1 mM CaCl$_2$, and 4 mM MgCl$_2$, bubbled with 5% CO$_2$, 95% O$_2$ at 22°C. The 1 mM CaCl$_2$ was replaced by 1 mM MgCl$_2$ to record mEPSCs and mIPSCs in 0 mM of Ca$^{2+}$. Whole-cell recordings were carried out at –60 mV for all EPSCs, including mEPSCs, evoked EPSCs, and sucrose-evoked responses. The holding potential was switched to 0 mV to record mIPSCs. The internal solution contained 105 mM CH$_3$O$_3$SCs, 10 mM CsCl, 15 mM CsF, 4 mM MgCl$_2$, 5 mM EGTA, 0.25 mM CaCl$_2$, 10 mM HEPES, and 4 mM Na$_2$ATP, adjusted to pH 7.2 using CsOH. Stimulus-evoked EPSCs were stimulated by placing a borosilicate pipette (5–10 μm) near the ventral nerve cord (one muscle distance from the recording pipette) and applying a 0.4 ms, 85 μA square pulse using a stimulus current generator (WPI).

### Fluorescence imaging

Animals were immobilized on 2% agarose pads with 30 mM levamisole. Fluorescence imaging was performed on a spinning-disk confocal system (3i Yokogawa W1 SDC) controlled by Slidebook 6.0 software. Animals were imaged with an Olympus 100×1.4 NA Plan-Apochromat objective. Z series of optical sections were acquired at 0.13 μm steps. Images were deconvolved with Huygens Professional version 16.10 (Scientific Volume Imaging, The Netherlands) and then processed to yield maximum intensity projections using ImageJ 1.54h (Wayne Rasband, National Institutes of Health) (*Schneider et al., 2012*).

## Quantification and statistical analysis

### Data acquisition and analysis

All electrophysiological data were obtained using a HEKA EPC10 double amplifier (HEKA Elektronik) filtered at 2 kHz, and analyzed with open-source scripts developed by Eugene Mosharov (http://sulzerlab.org/Quanta_Analysis_8_20.ipf) in Igor Pro 7 (WaveMetrics). All imaging data were analyzed in ImageJ software. Each set of data represents the mean ± SEM of an indicated number (n) of animals. To analyze mEPSCs and mIPSCs, a 4 pA peak threshold was preset, above which release events are clearly distinguished from background noise. The analyzed results were re-checked by eye to ensure that the release events were accurately selected.

## Statistical analysis

All data were statistically analyzed in Prism 9 software. Normality distribution of the data was determined by the D'Agostino-Pearson normality test. When the data followed a normal distribution, an unpaired Student's t-test (two-tailed) or one-way ANOVA was used to evaluate the statistical significance. In other cases, a Mann-Whitney test or one-way ANOVA following Kruskal-Wallis test was used. A summary of all electrophysiological data is provided in *Supplementary file 1*, with the results presented as mean ± SEM.

## Acknowledgements

We thank the *C. elegans* Genetics Stock Center for strains and reagents. We thank members of the Hu lab. This work was supported by a National Health and Medical Research Council Project grant (APP1122351 to ZH), a CityU startup fund (9610647 to ZH), and a National Institutes of Health research grant (R56NS128048 to JR and ZH).

## Additional information

### Funding

| Funder | Grant reference number | Author |
| --- | --- | --- |
| National Health and Medical Research Council | APP1122351 | Zhitao Hu |
| National Institutes of Health | R56NS128048 | Zhitao Hu |
| City University of Hong Kong | 9610647 | Zhitao Hu |

The funders had no role in study design, data collection and interpretation, or the decision to submit the work for publication.

### Author contributions

Haowen Liu, Lei Li, Jingyao Xia, Validation, Investigation; Jiafan Wang, Jiayi Hu, Xiaochun Yu, Jing Tang, Investigation; Huisheng Liu, Cong Ma, Writing – review and editing; Xiaofei Yang, Validation, Writing – review and editing; Lijun Kang, Investigation, Writing – review and editing; Zhitao Hu, Supervision, Funding acquisition, Writing – original draft, Project administration, Writing – review and editing

### Author ORCIDs

Haowen Liu ⓘ https://orcid.org/0000-0001-8498-290X
Cong Ma ⓘ https://orcid.org/0000-0002-7814-0500
Lijun Kang ⓘ https://orcid.org/0000-0001-9939-5134
Zhitao Hu ⓘ https://orcid.org/0000-0002-2948-3339

Joint public review: https://doi.org/10.7554/eLife.105199.3.sa1
Author response https://doi.org/10.7554/eLife.105199.3.sa2

## Additional files

### Supplementary files

Supplementary file 1. Summary of all electrophysiology data in this study. The electrophysiology data from all strains used in this study are shown in the table (by mean ± SEM).

MDAR checklist

### Data availability

This study did not generate/analyze any code. The original electrophysiology data have been deposited in Dryad (https://doi.org/10.5061/dryad.2z34tmpxk).

The following dataset was generated:

| Author(s) | Year | Dataset title | Dataset URL | Database and Identifier |
|---|---|---|---|---|
| Liu H, Li L, Wang J, Hu J, Xia J, Yu X, Tang J, Liu H, Yang X, Ma C, Kang L, Hu Z | 2025 | Data from "Mechanisms that regulate the C1-C2B mutual inhibition control functional switch of UNC-13" | https://doi.org/10.5061/dryad.2z34tmpxk | Dryad Digital Repository, 10.5061/dryad.2z34tmpxk |

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
