## [Editor Report · eLife Assessment]

This **important** study by Liu et al. presents a comprehensive structure-function analysis of the presynaptic protein UNC-13, leading to new insights into how its distinct domains control neurotransmitter release. The methods, data, and analyses are **convincing**, and the genetic and electrophysiological approaches support many of their conclusions. The work will be of interest to neuroscientists studying synaptic transmission, as it provides a foundation for future mechanistic studies of Munc13/UNC-13 family proteins.

---

## [Referee Report · Joint public review]

Summary:

In this manuscript, the authors investigate how different domains of the presynaptic protein UNC-13 regulate synaptic vesicle release in the nematode *C. elegans*. By generating numerous point mutations and domain deletions, they propose that two membrane-binding domains (C1 and C2B) can exhibit "mutual inhibition," enabling either domain to enhance or restrain transmission depending on its conformation. The authors also explore additional N-terminal regions, suggesting that these domains may modulate both miniature and evoked synaptic responses. From their electrophysiological data, they present a "functional switch" model in which UNC-13 potentially toggles between a basal state and a gain-of-function state, though the physiological basis for this switch remains partly speculative.

Strengths:

(1) The authors conduct a thorough exploration of how mutations in the C1, C2B, and other regulatory domains affect synaptic transmission. This includes single, double, and triple mutations, as well as domain truncations, yielding a large, informative dataset.

(2) The study includes systematically measure both spontaneous and evoked synaptic currents at neuromuscular junctions, under various experimental conditions (e.g., different Ca²⁺ levels), which strengthens the reliability of their functional conclusions.

(3) Findings that different domain disruptions produce distinct effects on mEPSCs, mIPSCs, and evoked EPSCs suggest UNC-13 may adopt an elevated functional state to regulate synaptic transmission.

---

## [Author Response]

The following is the authors’ response to the original reviews.

Overview:

We appreciate all the constructive comments from the reviewer and the reviewing editor, as their suggestions have significantly improved our manuscript. In response to their comments, we have made several key revisions: First, we have performed new colocalization analyses between the active zone marker UNC-10::GFP and all UNC-13L variants (UNC13L, UNC-13L^HK^, UNC-13L^D1-5N^, and UNC-13L^HK+D1-5N^, all tagged with mApple). These results confirm that the mutations do not affect synaptic localization. Second, we have provided a clearer explanation of the “gain-of-function” term used in this study, emphasizing that it reflects an increased SV release due to C1-C2B module dysfunction rather than a single mechanistic state. Third, we have expanded the discussion on the physiological implications of the C1-C2B model, particularly its role in regulating synaptic transmission under varying neuronal activity conditions. Finally, to improve clarity and focus, we have removed unnecessary speculative discussions, ensuring that the revised manuscript centers on the most relevant findings.

We have reorganized the manuscript to incorporate these new results into the figures and text. Full responses to all reviewer comments are provided below. We hope that the reviewer and the editor find these revisions satisfactory and that our manuscript is now suitable for publication in *eLife*.

**Joint Public Review:**
Summary:In this manuscript, the authors investigate how different domains of the presynaptic protein UNC-13 regulate synaptic vesicle release in the nematode *C. elegans.* By generating numerous point mutations and domain deletions, they propose that two membrane-binding domains (C1 and C2B) can exhibit "mutual inhibition," enabling either domain to enhance or restrain transmission depending on its conformation. The authors also explore additional Nterminal regions, suggesting that these domains may modulate both miniature and evoked synaptic responses. From their electrophysiological data, they present a "functional switch" model in which UNC-13 potentially toggles between a basal state and a gain-of-function state, though the physiological basis for this switch remains partly speculative.Strengths:(1) The authors conduct a thorough exploration of how mutations in the C1, C2B, and other regulatory domains affect synaptic transmission. This includes single, double, and triple mutations, as well as domain truncations, yielding a large, informative dataset.(2) The study includes systematically measuring both spontaneous and evoked synaptic currents at neuromuscular junctions, under various experimental conditions (e.g., different Ca²⁺ levels), which strengthens the reliability of their functional conclusions.(3) Findings that different domain disruptions produce distinct effects on mEPSCs, mIPSCs, and evoked EPSCs suggest UNC-13 may adopt an elevated functional state to regulate synaptic transmission.Weaknesses:It remains unclear whether the various domain alterations truly converge on a single "gain-offunction" state or instead represent multiple pathways for enhancing UNC-13 activity. Different mutations selectively affect spontaneous or evoked release, suggesting that each variant may not share the same underlying mechanism. Moreover, many conclusions rely on combining domain deletions or point mutations, yet the electrophysiological data show distinct outcomes across EPSCs, IPSCs, mini, and evoked responses. This raises questions about whether these manipulations all act on the same pathway and whether their observed additivity or suppression genuinely reflects a single mechanistic process. A unifying model-or at least a clearer explanation of why the authors infer one mechanistic state across different domain manipulations would strengthen the paper's conclusions.

We appreciate the comment and understand the potential confusion regarding the use of the term "gain-of-function" in the manuscript. To clarify, the gain-of-function state described in this study does not refer to a single specific mechanistic change in UNC-13 but rather to a high synaptic vesicle (SV) release state achieved by disrupting the C1-C2B module - either through dysfunction of the C1 domain or the C2B domain (as seen with the HK and DN mutations).

Our findings support a "seesaw" model in which the C1 and C2B domains maintain a dynamic balance in their interaction with the plasma membrane, binding to DAG and PIP2. This balance may increase the energy barrier for SV release, preventing excessive neurotransmitter release under basal conditions. However, the C1-C2B toggle may be disrupted by high neuronal activity and act in an unbalanced state, thereby enhancing synaptic transmission (i.e., the gain-of-function state). To address these concerns, we have provided a clearer explanation of this functional switch in the revised version of the manuscript (page 27).

Regarding the differences between spontaneous and evoked neurotransmitter release, our previous studies have revealed that these two forms of release do not always respond similarly to various *unc-13* mutations. This is a common phenomenon observed in other synaptic protein mutants, including synaptotagmin, tomosyn, and complexin, which indicates distinct yet partially overlapping regulatory mechanisms. Our model is well supported by most of the electrophysiological results from HK, DN, and HK+DN mutations across different *unc-13* isoforms (UNC-13L, UNC-13S, UNC-13R, UNC-13ΔC2A, UNC-13ΔX). The main exception is that in UNC-13ΔX^HK+DN^ mutants, the changes in mEPSCs and mIPSCs differ from those observed in evoked EPSCs. This suggests that the mechanisms regulating the functional switch of *unc-13* may differ slightly between spontaneous and evoked release. Since the *X* region of *unc-13* and Munc13 remains largely uncharacterized, our findings provide intriguing insights into its potential functional role.

The manuscript proposes that UNC-13 toggles from a basal to a "gain-of-function" state under normal synaptic activity. However, it does not address when or how this switch might occur in vivo, since it is demonstrated principally via artificial mutations. Providing direct evidence or additional discussion of such switching under physiological conditions would be particularly informative.What is the physiological significance of the proposed gain-of-function state? The data suggest that certain mutants (e.g., HK+D1-5N) lacking the gain-of-function state can still support synaptic transmission at wild-type levels. How do the authors reconcile this with the idea that the gain-of-function state plays a critical role at the synapse?

We appreciate these comments. While our model is mainly based on the dysfunction of the C1-C2B module (through HK and DN mutations), it provides a potential physiological framework for understanding how the structural balance of C1-C2B relates to the variability of synaptic transmission in the nervous system. In the CNS, synaptic transmission is highly variable, and the temporal pattern of the presynaptic activity may require dynamic switching of the fusion machinery, including UNC-13, between different functional modes, thereby triggering synaptic transmission at various levels. Our model suggests that under conditions of high neuronal activity, the C1-C2B module may transition from a balanced to an unbalanced state (gain-of-function state), thereby enhancing synaptic transmission.

Regarding the physiological significance of the gain-of-function state, we acknowledge that certain mutants (e.g., HK+D1-5N) lacking this state can still support wild-type levels of synaptic transmission. This observation suggests that the gain-of-function state may not be strictly required for baseline synaptic function but rather plays a modulatory role under specific conditions, such as heightened neuronal activity or synaptic plasticity. Further investigations will be needed to determine the precise in vivo triggers and functional consequences of this switch under physiological conditions. Moreover, we will focus on several linker regions (between C1 and C2B, C2B and MUN) to investigate their potential roles in regulating synaptic transmission and their broader functional significance in UNC-13 dynamics.

The authors determined the fluorescence intensity of mApple-tagged UNC-13 variants (Figure 1J-K and Figure 7J-K), finding no significant changes compared to the wild-type. However, a more detailed analysis of the density or distribution of fluorescent puncta in axons could clarify whether certain mutations alter the localization of UNC-13 at synapses. Demonstrating colocalization with wild-type UNC-13 (or another presynaptic marker) would help rule out mislocalization effects.

We appreciate the comment. In response, we have included a more detailed analysis of the synaptic localization of both wild-type and mutated UNC-13L in the revised manuscript. Our data show that in all scenarios, UNC-13 proteins exhibit strong colocalization with the active zone marker UNC-10::GFP (Figure 1L). Along with the fluorescence intensity data in Figure 1J, our findings indicate that the C1 and C2B mutations do not affect the expression level or the localization of UNC-13 at synapses. These results have been incorporated into the revised manuscript (page 8) and in Figure 1L.

The study mainly relies on extrachromosomal transgenes, which can show variable copy numbers and expression levels among individual worm strains. This variability might complicate interpretation, as differences in expression could mask or exaggerate certain phenotypes.

We agree that the expression levels of synaptic proteins can influence synaptic transmission levels. However, given the large number of mutations and truncations employed in this study, generating single-copy rescue lines for all transgenic strains would be a significant undertaking. On average, we need to microinject 50-100 worms to obtain one single-copy line, whereas injecting only 5-10 worms allows us to generate at least three independent extrachromosomal arrays. Based on our previous work, we found that the synaptic transmission levels are comparable between various extrachromosomal rescue arrays of *unc13* and their single-copy rescue lines (e.g., UNC-13L, UNC-13S, UNC-13R, UNC-13ΔC2A, UNC-13ΔC2B, etc.). In future studies, we aim to use single-copy expression or CRISPRbased methods to induce deletions or mutations in various synaptic proteins.

Finally, the discussion is somewhat diffused. Streamlining the text to focus on the most direct connections would help readers pinpoint the key conclusions and open questions.

We appreciate the comment. As suggested, we have refined the discussion section. Specifically, we have removed the last part of the discussion (Functional roles of the linkers in UNC-13).

**Recommendations for the authors:**

**Reviewer #1 (Recommendations for the authors):**
(1) Clarify the "Gain-of-Function" State. Provide stronger justification or explicit discussion of whether all manipulations that enhance SV release truly correspond to the same mechanistic state or if multiple conformational states might be at play.

The “gain-of-function” state in this manuscript refers to a specific conformational status of UNC-13 that enhances synaptic vesicle (SV) release probability (both spontaneous and evoked) as a result of mutations (HK and DN) in the C1 and C2B domains. This effect is observed across multiple UNC-13 isoforms, including UNC-13L, UNC-13S, and UNC-13R. Prior studies from our group and others have demonstrated that C1 and C2B exhibit conserved functions in regulating synaptic transmission (Li et al., 2019, Cell Reports; Liu et al., 2021, Cell Reports; Michelassi et al., 2017, Neuron), supporting the idea that these domains share a common mechanism for modulating SV release. Given that C1 and C2B act as a functional unit (Michelassi et al., 2017, Neuron; and this study), we define all synaptic states induced by the dysfunction of these two domains as the "gain-of-function" mode.

However, it is important to note that this classification does not apply to high-release probability states induced by mutations in other domains.

The concept of a gain-of-function state due to C1 and C2B dysfunction has been previously proposed in studies of Munc13. Basu et al. (2007, Journal of Neuroscience) demonstrated that the H567K mutation in Munc13-1 C1 increases both spontaneous and evoked release probability, leading to a gain-of-function mode. Similarly, work from the Südhof group showed that KW and DN mutations in Munc13-1 C2B also enhance release probability, thereby inducing a gain-of-function state (Shin et al., 2010, Nature Structural & Molecular Biology). Our recent findings further support this idea, showing that UNC-13 C2B D3,4N (Li et al., 2019, Cell Reports; Liu et al., 2021, Cell Reports; Michelassi et al., 2017, Neuron) and the newly identified D1-5N mutation (this study) significantly elevate SV release, consistent with the D1,2N mutations reported by Shin et al.

Overall, our study integrates and extends previous findings, providing strong evidence that the C1 and C2B domains function as a regulatory switch between a basal physiological mode, a gain-of-function mode (enhanced release), and a loss-of-function mode (impaired release). This framework advances our understanding of how C1 and C2B dysfunction affects synaptic transmission and plasticity.

(2) Add comparisons to wild-type UNC-13L: When presenting data for deletions/mutants as "controls," include a visual reference (e.g., dashed line in figures) showing wild-type UNC13L levels. This will help readers see whether each construct is above or below the normal activity baseline.

As suggested, a dashed line showing the level of UNC-13L has been added to the bar graphs of all evoked EPSCs. The functional switch model is well supported by the results of the evoked EPSCs.

(3) Mutant and wild-type UNC-13 colocalization analysis: Demonstrating whether each mutant localizes robustly to synapses, in comparison to wild-type UNC-13, would bolster the interpretation of electrophysiological changes. If the authors have these data, adding them would address the possibility of mislocalization.

We agree with the reviewer that there would be value to address the possibility of mislocalization. However, in our experience working with UNC-13 mutant colocalization, we have found that neither deleting the X, C1 and C2B domains in UNC-13L nor deleting C1 and C2B domain in UNC-13MR or UNC-13R altered the synaptic colocalization with the active zone protein UNC-10/RIM (Li 2019, Liu 2021), suggesting that C1 and C2B domains in UNC-13 are not involved in the regulation of protein localization. Thus, the mutations in the C1 and C2B domains are unlikely leading to protein mislocalization in the synaptic region.

(4) If possible, adding analysis using single-copy transgenes to confirm that extrachromosomal array expression variability does not qualitatively change the conclusions.

We strongly agree with the reviewer that single-copy transgenes would provide more stable protein expression levels and further consolidate our conclusions. However, several factors give us confidence that the extrachromosomal array rescue approach does not introduce significant variability in our results: First, our prior research has shown that SV release levels are generally comparable between extrachromosomal arrays carrying various *unc13* transgenes and their corresponding single-copy rescue lines (e.g., UNC-13L, UNC-13S, UNC-13R, UNC-13ΔC2A, and UNC-13ΔC2B). Second, the major conclusions in this study are drawn from highly consistent and robust changes in SV release between different rescue lines (e.g., UNC-13L^HK+DN^ vs UNC-13L^DN^; UNC-13S^HK+DN^ vs UNC-13S^HK^ or UNC-13S^DN^). Third, our imaging data indicate that the protein levels are indistinguishable between different *unc-13* rescue arrays carrying C1 and C2B mutations, further supporting the validity of our findings.

Additionally, due to our recent relocation to a new institute, we are still in the process of setting up our microinjection system. Generating single-copy transgenes for all the extrachromosomal arrays used in this study would require significant time. We appreciate the reviewer’s understanding of our current situation. For our future studies regarding *unc-13* and other synaptic proteins, we will prefer to use single-copy expression rather than extrachromosomal arrays.

(5) Reduce the length and speculation in the Discussion. A concise discussion that focuses on the most direct implications of the present findings will help improve the readability of this paper.

We appreciate the comment. As suggested, we have refined the discussion section.

Specifically, the last part of the discussion (Functional roles of the linkers in UNC-13) was removed.

(6) Minor formatting detail: In Figure 5C (left panel), adjust the y-axis label to ensure it aligns properly and improves clarity.

We appreciate the reviewer’s suggestion and have adjusted the y-axis label accordingly in the revised version (see revised Figure 5).